# Structural basis of malaria parasite phenylalanine tRNA-synthetase inhibition by bicyclic azetidines

Manmohan Sharma[1,2,9], Nipun Malhotra[1,9], Manickam Yogavel[1,9], Karl Harlos [3], Bruno Melillo [4,5], Eamon Comer[4], Arthur Gonse[4], Suhel Parvez [2], Branko Mitasev[6], Francis G. Fang[6], Stuart L. Schreiber [4,7] & Amit Sharma[1,8 ✉]

The inhibition of *Plasmodium* cytosolic phenylalanine tRNA-synthetase (cFRS) by a novel series of bicyclic azetidines has shown the potential to prevent malaria transmission, provide prophylaxis, and offer single-dose cure in animal models of malaria. To date, however, the molecular basis of *Plasmodium* cFRS inhibition by bicyclic azetidines has remained unknown. Here, we present structural and biochemical evidence that bicyclic azetidines are competitive inhibitors of L-Phe, one of three substrates required for the cFRS-catalyzed aminoacylation reaction that underpins protein synthesis in the parasite. Critically, our co-crystal structure of a *Pvc*FRS-BRD1389 complex shows that the bicyclic azetidine ligand binds to two distinct sub-sites within the *Pvc*FRS catalytic site. The ligand occupies the L-Phe site along with an auxiliary cavity and traverses past the ATP binding site. Given that BRD1389 recognition residues are conserved amongst apicomplexan FRSs, this work lays a structural framework for the development of drugs against both *Plasmodium* and related apicomplexans.

[1] Molecular Medicine, Structural Parasitology Group, International Centre for Genetic Engineering and Biotechnology, Aruna Asaf Ali Marg, New Delhi 110067, India. [2] Department of Medical Elementology and Toxicology, School of Chemical and Life Sciences, Jamia Hamdard, New Delhi 110062, India. [3] Division of Structural Biology, Welcome Centre for Human Genetics, University of Oxford, Oxford OX3 7BN, England. [4] Chemical Biology and Therapeutics Science Program, Broad Institute of Harvard and MIT, 415 Main Street, Cambridge, MA 02142, USA. [5] Department of Chemistry, The Scripps Research Institute, 10550 North Torrey Pines Road, La Jolla, CA 92037, USA. [6] Eisai Inc., 35 Cambridgepark Drive Suite 200, Cambridge, MA 02140, USA. [7] Department of Chemistry and Chemical Biology, Harvard University, 12 Oxford Street, Cambridge, MA 02138, USA. [8] National Institute of Malarial Research, Sector 8 Dwarka, New Delhi 110077, India. [9] These authors contributed equally: Manmohan Sharma, Nipun Malhotra, Manickam Yogavel. ✉email: directornimr@gmail.com

Malaria, caused by apicomplexan parasites of the genus *Plasmodium*, presents a formidable global health challenge mainly due to the emergence of parasite strains that are resistant to front-line drugs[1,2]. It is therefore necessary to discover and validate new drug targets as well as compounds whose efficacy is unaffected by mechanisms of resistance to traditional antimalarials[2,3]. Ideally, an antimalarial development candidate should have a new mechanism of action (MoA) with rapid asexual blood-stage parasite reduction, and activity against all stages of the parasite lifecycle in the human host[2]. Recently, a series of small molecules based on a bicyclic azetidine core have been discovered that exhibit multistage antimalarial activity and can achieve single-dose cures in a mouse model of malaria[2]. It was demonstrated that these compounds exert their antimalarial activity via inhibition of *Plasmodium falciparum* cytosolic phenylalanyl-tRNA synthetase (cFRS)[2], an enzyme essential for protein synthesis. The *Pf*cFRS was indeed validated both genetically and biochemically as a drug target for the bicyclic azetidine series of molecules[2], setting the stage for further drug discovery efforts. Aminoacyl-tRNA synthetases (aaRSs) activate amino acids as aminoacyl adenylates (AA-AMP), and enable their relay to the 3′-ends of cognate tRNAs as feed for ribosomes[3,4]. Inhibition of aaRSs therefore results in the interruption of cell growth and ultimately in cell death[3]. Of note, *Plasmodium* aaRSs other than FRS have also recently become the focus of antimalarial development efforts[3].

As previously shown for *Plasmodiam falciparum*, the parasite has three protein translation compartments: in the cytoplasm, apicoplast and in the mitochondria where FRSs reside to feed charged tRNAs into ribosomal-based protein synthesis[4]. In both *P. falciparum* and *P. vivax*, FRSs exist as heterodimers of alpha (α) and beta (β) subunits that further dimerize to form a complex of (αβ)$_2$[4]. This hetero-tetrameric organization of cFRSs is conserved but with significant differences in the chain lengths and functions of α and β subunits[5]. The FRS α subunit contains the active site and catalyzes the two-step aminoacylation reaction, while the main functions of the β subunit are to recognize the anticodon region of tRNA and to edit mischarged tRNA molecules with isosteric amino acids such as tyrosine[5]. The cFRSs are highly conserved and exhibit high sequence identity amongst *Plasmodium* species suggesting that FRS from all five parasites causative of human malaria can be targeted by a single chemical series. In this work, we reveal the biochemical and structural basis of inhibition of *Plasmodium* cFRS via the x-ray co-crystal structure of *Pvc*FRS with BRD1389, a potent antimalarial from the bicyclic azetidine chemical series.

## Results

### BRD1389 binds cFRS selectively and inhibits aminoacylation via L-Phe competition.
To determine the mode of inhibition of cFRS by bicyclic azetidines, we turned to BRD1389 (Fig. 1a, Supplementary Fig. 1, 2), a series analogue with high in vitro potency in the growth inhibition assay (*Pf*3D7 and *Pf*K1 EC$_{50}$ are 13 and 12 nM respectively), potent abrogation of parasite FRS enzyme activity, and high selectivity index over the human orthologue (Table 1, Fig. 1b, Supplementary Fig. 3c, d). We measured the enzymatic activity of purified heterodimeric *Pf*cFRS using malachite green-based aminoacylation assays with substrates L-Phe and ATP[6]. Specifically, in order to understand *Pf*cFRS kinetics, enzyme activity was evaluated at a fixed saturating concentration of one substrate (L-Phe or ATP) and varying concentrations of the other at different inhibitor concentrations. The experimental data were then assessed for modes of competitive, mixed, or uncompetitive inhibition. We first performed this study at a constant concentration of ATP with increasing

concentrations of L-Phe. These data indicate that BRD1389 displays a mode of competitive inhibition vis-a-vis L-Phe binding (Fig. 1c) as fitting to a global competitive inhibition model resulted in $K_i$ of $6 \pm 2$ nM for the compound. These results are consistent with previously reported data on in vitro parasite growth inhibition by a closely related bicyclic azetidine (BRD1095) which showed higher EC$_{50}$ values with increasing concentrations of L-Phe[2]. We then repeated the mode of inhibition study using ATP as the variable-concentration substrate. The data were then fit to the modified high-substrate inhibition Michaelis–Menten equation (for the corresponding double-reciprocal plot, see Fig. 1d), consistent with a non-competitive inhibition model, where BRD1389 gave $K_i$ value of $10 \pm 5$ nM. Taken together, these results suggest that BRD1389 preferentially binds to the free enzyme via competition of L-Phe. This mechanism of action is unique when compared to other promising antimalarial aaRS inhibitors of cladosporin class[7–9] (it mimics adenosine) or the halofuginone scaffold (requires ATP for tight binding)[10–12]. Additionally, in order to understand the degree of selectivity, binding kinetics were measured using *Pf*cFRS and human cytosolic FRS (*Hsc*FRS). The data exhibits quite distinct binding affinities as well as binding kinetics (BRD1389 K$_d$: *Pf*cFRS: 4 nM; *Hsc*FRS: 16 μM) when fit to a 1:1 binding model of nonlinear regression (specific binding) (Supplementary Fig. 3c, d).

### Overall structure of *Pvc*FRS.
Toward unravelling the structural basis of the protein–inhibitor interaction, we crystallized the *Pvc*FRS-BRD1389 complex and obtained crystals that diffracted to 3 Å. The structure was solved by molecular replacement (MR) using *Hsc*FRS as a template (PDB: "3L4G [https://www.rcsb.org/structure/3l4g]"). The *Pvc*FRS-BRD1389 crystals belong to orthorhombic space group P2$_1$2$_1$2 with one heterodimer (αβ) per asymmetric unit. The (αβ)$_2$ biological heterotetrametric assembly is completed via the crystallographic two-fold axis along c. The *Pvc*FRS (αβ)$_2$ assembly is consistent with the size exclusion chromatography profile of purified protein, where it elutes at a size of ~298 kDa (Supplementary Fig. 3a). Our final *Pvc*FRS-BRD1389 atomic model contains 905 residues- 299:α subunit, 606:β subunit, one Mg$^{2+}$, and one ligand (BRD1389). The N-terminal domain (residues 1–270) of the α subunit was absent in the crystal and no electron density was observed for it. This was further verified by crystal packing analysis of *Pvc*FRS-BRD1389 complex and via SDS-PAGE of crystallised protein (Supplementary Fig. 4). The overall fold and organization of α and β subunit is very similar to that of the human orthologue (*Hsc*FRS, Fig. 2a, b) with the exception that the *Pvc*FRS β subunit lacks an 18-residue-long fragment within its PB1 domain whereas it has two unique insertions (IL-1: 421–444, IL2: 535–551) in its B2 subdomain (Supplementary Fig. 5). Intriguingly, the association of β1 and β2 subdomains (of the β subunit) with α subunit is significantly different between the parasite and human enzymes (Fig. 1e–j, Supplementary Fig. 6a). In *Hsc*FRS, the α subunit is enclosed by β1 and β2 subdomains of the same β subunit, whereas in *Pvc*FRS β1 and β2′ of the β subunit is associated with the α subunit (Fig. 1j, i). This difference arises due to the shorter linker length between β1 and β2 of *Pvc*FRS when compared to the *Hsc*FRS (Fig. 1h, k) which has a three residue (384-TYT-386) insertion. Interestingly, this three-residue shorter linker is observed in all human malaria parasites (Fig. 1h, k) and is suggestive of domain swapping (Fig. 1g) in *Pvc*FRS to form a closed, functional hetero-tetrameric (αβ)$_2$ assembly.

### BRD1389 occupies the L-Phe site and an auxiliary site on *Plasmodium* cFRS.
During the refinement of *Pvc*FRS model, an

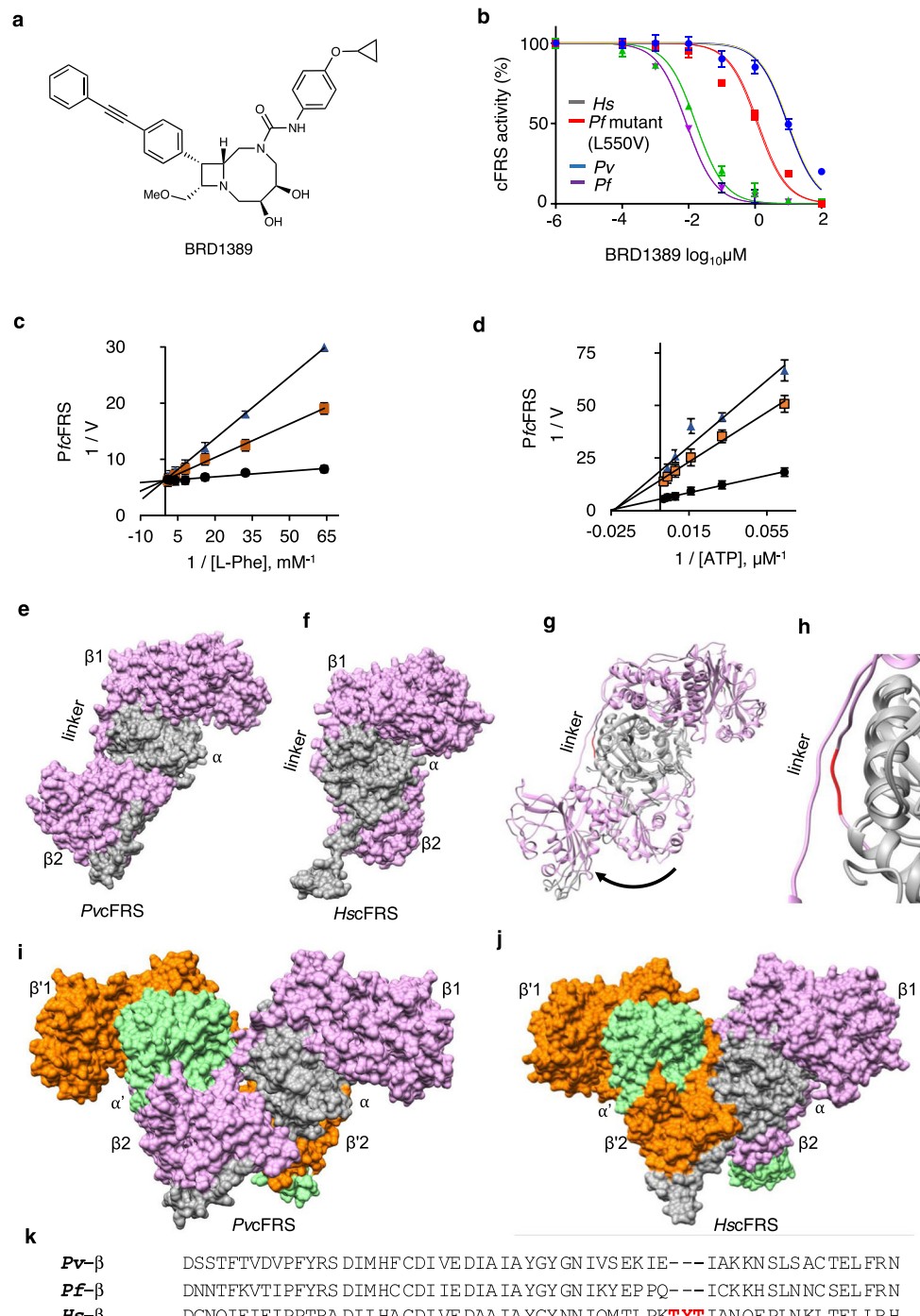

**Fig. 1 BRD1389 binds cFRS selectively and inhibits aminoacylation via L-Phe competition. a** Chemical structure of BRD1389. **b** Inhibition of the aminoacylation activity of *Pf* (Plasmodium falciparum), *Pv* (Plasmodium vivax), *Hs* (Homo sapiens) and *Pf* mutant (L550V) cFRS enzymes by BRD1389. These assays were performed at concentrations ranging from 100 μM to 0.1 nM and the $IC_{50}$ values were calculated by non-linear regression. Data are shown as mean ± SD ($n = 3$ independent experiments). **c, d** Mode of BRD1389 inhibition. BRD1389 is a competitive inhibitor of *Pf*cFRS with respect to L-Phe ($K_i = 6$ nM) while it is likely a non-competitive inhibitor with respect to ATP ($K_i = 10$ nM). Lineweaver-Burk plots were obtained at a saturating concentration of either ATP (500 μM) or L-Phe (1000 μM) with varying concentrations of other substrate: L-Phe (1000 – 15.6 μM) or ATP (500–15.6 μM) and inhibitor BRD1389 (1 × $IC_{50}$ (blue triangle), 0.5 × $IC_{50}$ (yellow square), and 0 × $IC_{50}$ (black circle) Data are shown as mean ± SD ($n = 3$ independent experiments). The error bars indicate standard deviation ($n = 3$). **e, f** Surface view of heterodimeric assembly (αβ) of *Pv*cFRS and *Hs*cFRS (α-subunit in grey and β-subunit in pink) structures. The subdomains of β1 and β2 of β-subunit are marked. **g** Superimposed structure of *Pv/Hs*cFRS displaying movement of subdomain β2 of β-subunit and hence the possible domain swap in the malaria enzyme. **h** Close-up view of β-subunit linker region indicating the three residue insertion (red) in *Hs*cFRS, which when absent may lead to domain swap. **i, j** Surface view of the heterotetrameric assembly (αβ)$_2$ of both *Pv* and *Hs*cFRS structures (α-subunit is in grey and β-subunit is in plum whereas the symmetry related α′ and β′ domains are shown in light green and orange respectively). **k** Portion of the sequence alignment showing the linker region between β1 and β2 subdomains of the β-subunit in *Pv*cFRS versus *Hs*cFRS. Source data are provided as a Source Data file.

**Table 1 BRD1389 potency and selectivity for malaria parasite cFRS over *Hsc*FRS.**

| Experiment | Organism | Values (nM) |
|---|---|---|
| EC$_{50}$ | | |
| | *Pf* (3D7) | 13 |
| | *Pf* (K1) | 12 |
| IC$_{50}$ (cFRS) | *Pf* | 12 ± 0.8 |
| | *Pv* | 25 ± 1 |
| | *Pf* mutant (L550V) | $(1.0 ± 0.05) \times 10^3$ |
| | *Hs* | $(12 ± 1.2) \times 10^3$ |
| K$_d$ (cFRS) | *Pf* | 4 |
| | *Hs* | $16 \times 10^3$ |

*Pf* Plasmodium falciparum, *Pv* Plasmodium vivax, *Hs* Homo sapiens

investigation of difference Fourier electron density (F$_o$-F$_c$) maps revealed BRD1389 to be bound at the enzyme active site (Fig. 2c, d). The bound ligand was subsequently verified by simulated annealing omit (SA-omit) map (Fig. 2d). Superposition of the *Pv*cFRS α subunit on *Hsc*FRS and *Thermus thermophilus* FRS (*Tt*FRS, PDB: "2IY5 [https://www.rcsb.org/structure/2IY5]") allowed the mapping of L-Phe, ATP and tRNA binding sites on these FRSs (Fig. 3c). Further analysis of the *Pv*cFRS:BRD1389 complex revealed that the diarylacetylene moiety of BRD1389 occupies the L-Phe site, its 4-cyclopropoxy phenyl resides in an auxiliary pocket, and the [6.2.0]-diazabicyclodecane group skirts the ATP site in *Pv*cFRS (Fig. 3d, Supplementary Fig. 7). The *Pv*cFRS:BRD1389 complex is stabilized mainly by hydrophobic and hydrophilic interactions at multiple sites that contribute towards recognition of the diazabicyclodecane core, the 4-cyclopropoxyphenyl and the diarylacetylene appendages (Fig. 3a).

**The active site of BRD1389-bound *Plasmodium* cFRS adopts unique conformations**. In this complex, open and close conformation for residue Arg548 (Fig. 4a, d) is noticeable. In particular, the flexing of Arg548 likely opens the entry point of auxiliary pocket for 4-cyclopropoxyphenyl accommodation, underscoring the induced-fit nature of BRD1389 interaction with *Pv*cFRS (Fig. 4d). In phenyladenylate-bound *Tt*FRS, both the guanidinium moiety of Arg321 and Phe216 (*Pv*cFRS; Phe455) provide stacking support to the adenine ring of ATP, whereas in *Pv*cFRS-BRD1389 the corresponding guanidinium (belonging to Arg548) is displaced away from the active site adopting an open conformation (Fig. 4a, d). This can be further investigated in L-Phe-bound *Hsc*FRS, where the corresponding Arg463 moves inward (i.e., towards the active site) adopting instead a closed conformation. Additionally, in the *Pv*cFRS:BRD1389 complex there are two noticeable, ordered loop conformations in proximity to BRD1389 binding site within the PA1 domain: (1) a left-hand outward displacement (open conformation) of residues 443–453 (loop 1, in ATP binding pocket), and (2) a left-hand inward movement (closed) of residues 507–515 (in auxiliary pocket) (Fig. 4a–c, Supplementary Fig. 6b, c). In *Pv*cFRS: BRD1389, loop 2 adopts a closed conformation akin to phenyladenylate-bound *Tt*FRS and unlike L-Phe-bound *Hsc*FRS (open conformation, Fig. 4c, Supplementary Fig. 6c), whereas loop 1 moves relative to both *Tt*FRS and *Hsc*FRS to achieve an open state (Fig. 4b, Supplementary Fig. 6b).

**Structural basis of BRD1389 recognition by *Pv*cFRS.** The diarylacetylene moiety of BRD1389 that occupies the L-Phe site is recognized via a bed of β-strand residues Ala541-Trp542-Gly543-Leu544 (Fig. 3b), while its edge phenyl ring is covered by Tyr497

where it provides an aromatic T-shaped π…π interaction at a distance of ~4.2 Å (Fig. 3b). In addition, the diarylacetylene moiety is surrounded by hydrophobic elements of Asn519, Gln457, and Glu459 (Fig. 3b). The urea moiety of BRD1389 is buried in a groove that is composed of Gly506, His508, Glu510, Lys512–513, Leu515, Ile552 and Pro549 (Figs. 3b, e and 4c). The urea/groove interactions are mainly stabilized by hydrophobic contacts, particularly from Pro549 and His508 while residues Val517, Ile483 and Pro549 provide sandwich support to the cyclopropoxy moiety (Figs. 3b and 4c). The urea component of BRD1389 forms hydrogen bonding interactions with main-chain *N*-atom of Ser545 (Fig. 3a, b). The above sets of extensive interactions position BRD1389 in an 'L' shaped conformation wherein its methoxy methyl group is surrounded by socket residues Arg443, Glu445, His451, and Phe455. Interestingly, the crystallographic pose of BRD1389 is close to the conformation that the inhibitor is predicted to adopt in aqueous solution (Supplementary Fig. 8 and Supplementary Table 2). This similarity highlights the importance of the three-dimensional shape and rigidity of the diazabicyclodecane scaffold in pre-orienting the molecular appendages for optimal target engagement. From overlaying the structure of *Pv*cFRS:BRD1389 with that of phenyladenylate-bound *Tt*FRS, it is apparent that the diazabicyclodecane core and its methoxymethyl extension partially brush past the adenine binding region of the canonical ATP binding site (Figs. 3d and 4b). Given the high binding affinity of BRD1389 for *Pv*cFRS (4 nM, Table 1), it is feasible that BRD1389 blocks the interaction of *Plasmodium* cFRS with L-Phe first and then with ATP. Indeed, upon incubation of *Pv*cFRS with high concentrations of both BRD1389 and an ATP analogue (the non-hydrolysable adenosine 5'-(β,γ-imido) triphosphate, i.e. AMPPNP) we observed only binding of BRD1389. This result further supports that BRD1389 binding may occlude ATP engagement. Strikingly, residues in *Pv*cFRS that recognize key ligand components (diazabicyclodecane core, 4-cyclopropoxy phenyl, methoxymethyl and diarylacetylene moieties) are conserved across the apicomplexan phyla, including in human-infecting parasites such as *Toxoplasma* and *Cryptosporidium* (Supplementary Fig. 9).

**Basis of selectivity and resistance-conferring mutations**. Next, to understand the structural basis of selective binding and inhibition by bicyclic azetidines of *Plasmodium* cFRS versus the human orthologue, we compared atomic structures of *Pv*cFRS:BRD1389 and *Hsc*FRS-L-Phe complexes focusing on residues located within 5 Å of the ligand site. Three variant residues *Pv*-V458/Hs-I373, *Pv*-Y480/Hs-F395, *Pv*-I483/Hs-L398 are located within an auxiliary pocket of *Pv*cFRS (Fig. 5a). This observation indicates that the selectivity of BRD1389 may arise from the terminal cyclopropyl ether (Fig. 5a). Significantly, all protein residues known to confer resistance to bicyclic azetidines upon mutation[2] are located within the α subunit of FRS, in proximity to the BRD1389 (Fig. 5b). None of these mutations are directly in the ATP, tRNA, or L-Phe sites (Fig. 5b, Supplementary Table 3). However, one significant mutation *Pv*cFRS-L544V (equivalent to *Pf*cFRS-L550V) that diminishes BRD1389 potency structurally underpins the [6.2.0]-diazabicyclodecane ring of BRD1389 (Table 1, Fig. 5b, d). When we reconstructed the *Pf*cFRS-L550V and tested its enzymatic fidelity, we observed a significant increase in K$_m$ value (10-fold) for the substrate L-Phe, but not for ATP, indicating that resistance to bicyclic azetidines may arise from a trade-off between parasite survival and the essential nature of cFRS enzymatic activity (Fig. 5c, d).

**Discussion**
Through a combination of biochemical and crystallographic studies, we have revealed the molecular underpinning of

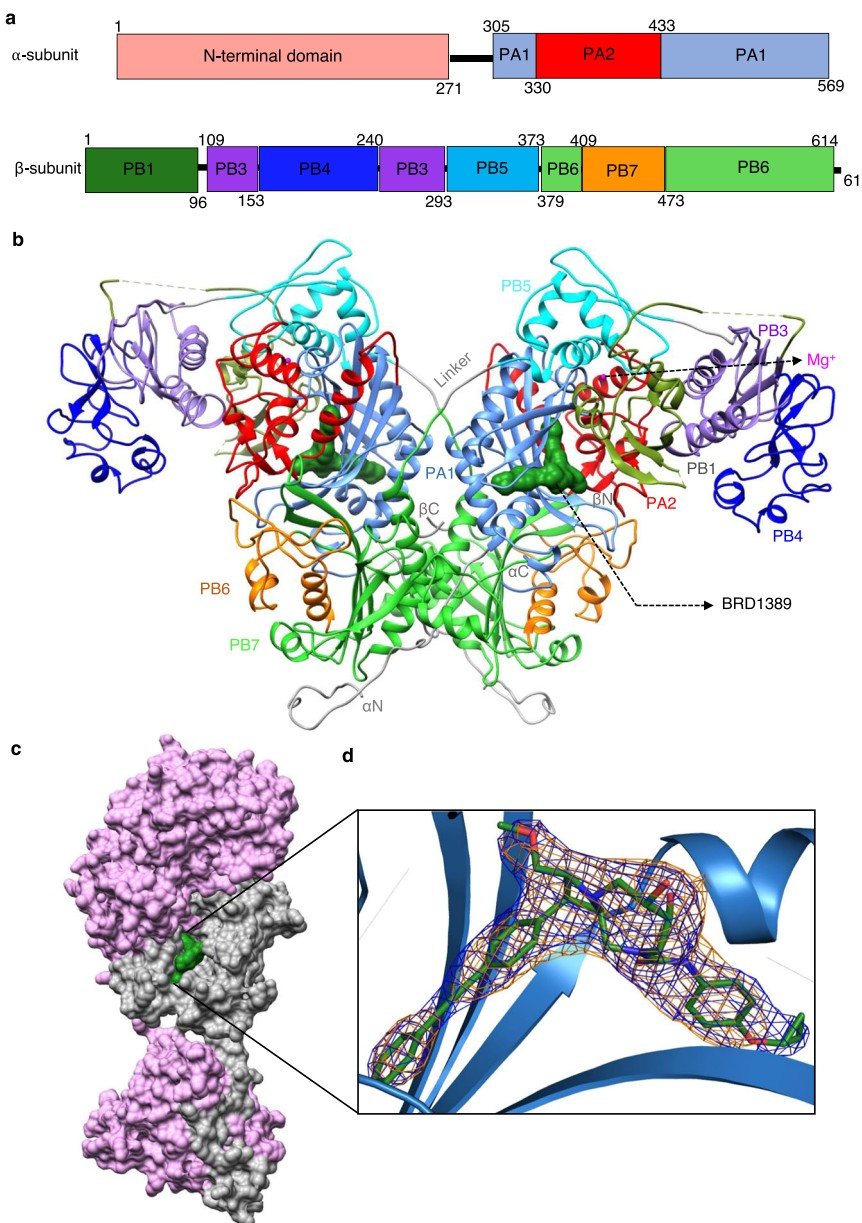

**Fig. 2 BRD1389 binds at the active site of *Plasmodium* cFRS. a** Structural organization of *Pvc*FRS α- and β-subunits. **b** Overall structure of *Pvc*FRS showing functional biological heterotetrametric assembly (αβ)$_2$ with two crystallographic heterodimers (αβ). The domain boundaries are labelled according to *Tt*FRS (PDB "2IY5 [https://www.rcsb.org/structure/2IY5]"). The α-subunit consists of two domains: PA1 (catalytic domain, CAT) and PA2 domain. The β-subunit consists of PB1, PB3 (editing domain), PB4, PB5 and PB6-B7 (catalytic-like, CAM). **c** Surface view of *Pvc*FRS:BRD1389 complex depicting α-subunit (grey), β-subunit (pink) and bound BRD1389 (green) in the α-subunit. **d** The composite simulated annealed omit (SA-omit) (orange) and the final 2F$_o$-F$_c$ (blue) maps are contoured at 1 σ levels for the bound BRD1389. The final 2F$_o$-F$_c$ (blue) map clearly shows the continuous electron density for the bound BRD1389.

*Plasmodium* cFRS inhibition by bicyclic azetidines. We have shown that BRD1389 inhibits parasite cFRS function by primarily blocking the binding of L-Phe in a competitive manner. The diphenylacetylene moiety of BRD1389 occupies the L-Phe binding site while the [6.2.0]-diazabicyclodecane core partially occludes the ATP binding region. The cyclopropoxyphenyl urea region of BRD1389, in turn, occupies an auxiliary pocket in *Pvc*FRS. Residue variations between the malaria parasite cFRS and the human orthologue in this region underpin the highly selective enzyme inhibition and parasite killing by bicyclic azetidines. Two classes of malaria parasite aaRS inhibitors have been structurally evaluated to date[3]. These act either as single-site

occupants (cladosporin, an adenosine mimic) or dual site engagers (halofuginone, a mimic of L-Pro and 3′ end of tRNA). As BRD1389 occupies both the L-Phe site and an auxiliary pocket within *Pvc*FRS, it represents a novel dual-site malaria parasite aaRS inhibitor (Fig. 6a, 6; right panel).

Overall, our mapping of protein regions and residues that contribute both to cFRS inhibitor selectivity and drug resistance provides a structural platform for designing the next generation of compounds with improved potency and safety profiles. Indeed, the enzyme-inhibitor structure also reveals how the underlying principles of the diversity-oriented synthesis (DOS)[13–15] library that yielded this inhibitor scaffold, i.e., inclusion of rigid bicyclic

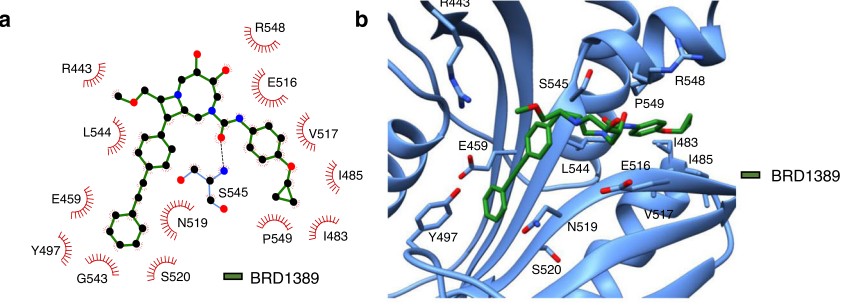

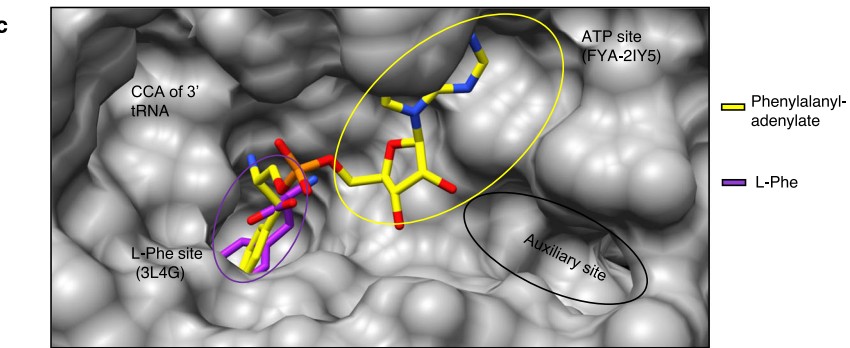

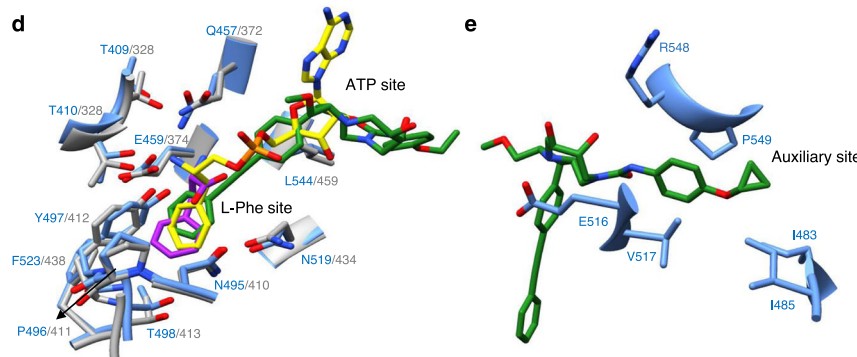

**Fig. 3 BRD1389 occupies the L-Phe site and an auxiliary site on *Plasmodium* cFRS. a** Two dimensional representation (Ligplot) of BRD1389 binding to *Pvc*FRS. BRD1389 is shown as ball-and-stick representation and interaction symbols of residues engaging in hydrophobic interactions with the ligand are highlighted in red. **b** Close-up view of bound BRD1389 in the active site of α-subunit of *Pvc*FRS. Labelled residues show hydrophobic interactions with BRD1389. **c** Surface view of the active site and pockets within L-Phe site (purple circle), ATP site (yellow circle) and auxiliary site (black circle). Superimposed structures of L-Phe (purple, PDB "3L4G [https://www.rcsb.org/structure/3l4g]") and phenylalanyl-adenylate (yellow, PDB "2IY5 [https://www.rcsb.org/structure/2IY5]") are depicted. **d** Close-up view of amino acid and ATP pocket of *Pvc*FRS (cornflower blue) that is superimposed on *Hsc*FRS-L-Phe (grey, PDB "3L4G [https://www.rcsb.org/structure/3l4g]"). **e** Close-up view of auxiliary site occupied by BRD1389. Auxiliary site residues are shown which are involved in protein-ligand hydrophobic interactions.

skeletons and multiple stereogenic elements—played a key role in accessing pockets within the enzyme that may have been inaccessible by compounds in classical libraries. Traditional libraries are replete with compounds that have a high percentage of atoms with sp² hybridization, leading to flatter architectures. BRD1389, which results from DOS, in contrast makes sharp turns and penetrates into deep pockets within *Pvc*FRS that are nearly at right angles.

This work may thus allow generation of compound libraries with tuneable drug-like properties that can focus on other apicomplexan-driven human diseases via targeting their FRSs. In particular, guided by structure, triple site inhibitors can now be developed that fully occupy the ATP site via chemical modifications of the [6.2.0]-diazabicyclodecane scaffold. More generally,

this work paves the way for novel drug development against malaria and, potentially, other diseases caused by apicomplexans such as toxoplasmosis and cryptosporidiosis.

## Methods

**Prediction of BRD1389 conformation**. Calculations were performed using Schrödinger Maestro Version 12.3.012, MMshare Version 4.9.012, Release 2020-1, Platform Windows-x64. Geometry optimization was performed in Jaguar[16] Version 10.7, Release 12, at ultrafine accuracy level, using density functional theory (B3LYP), the 6–31 G** basis set, and the Poisson–Boltzmann Finite (PBF) water solvation model. The crystallographic and computed structures were superposed by Maximum Common Substructure using the Schrödinger Maestro Superposition tool.

**Synthesis of BRD1389**. BRD1389 was prepared from known compound (8 *R*,9 *R*,10 *S*,*Z*)-9-(4-bromophenyl)-6-((4-nitrophenyl) sulfonyl)-10-((trityloxy)methyl)-

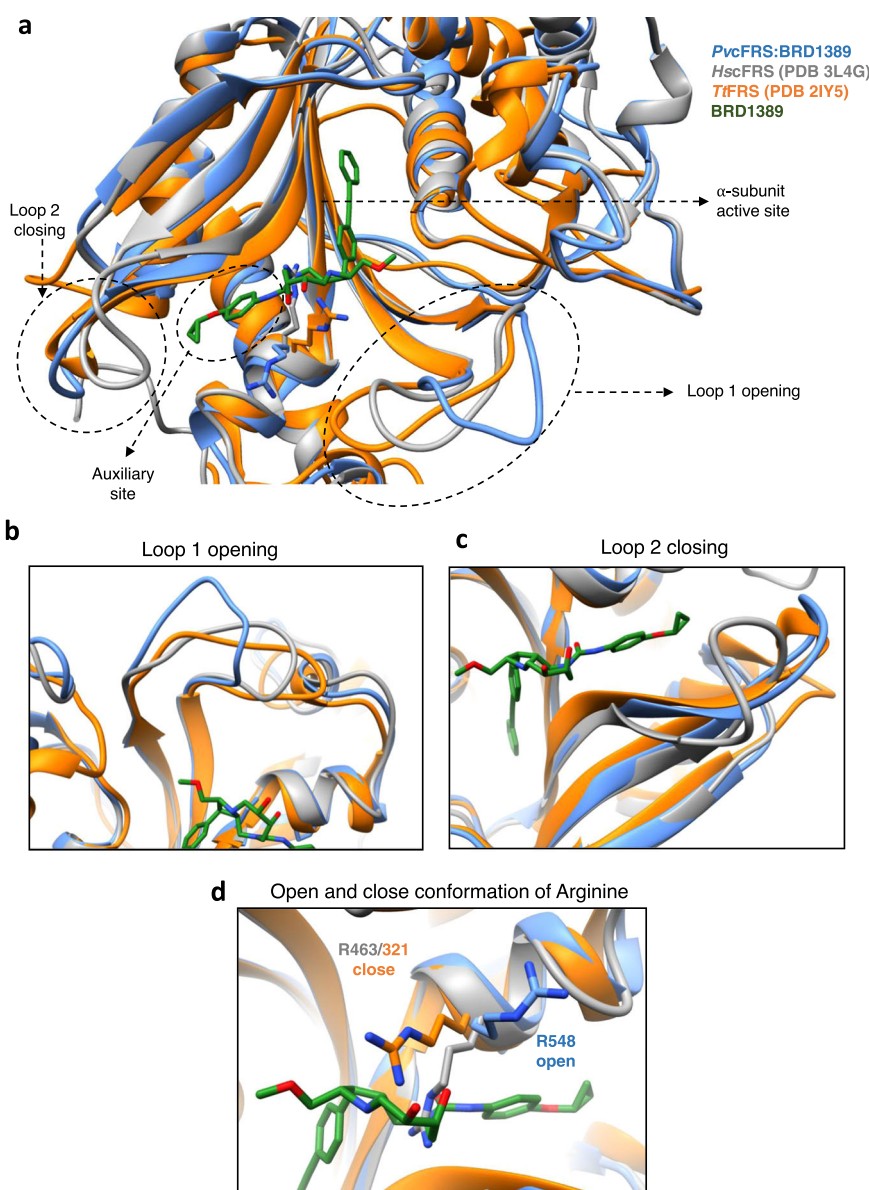

**Fig. 4 BRD1389-bound *Plasmodium* cFRS adopts unique conformations. a** Superimposition of *Hsc*FRS (grey; PDB "3L4G [https://www.rcsb.org/structure/3l4g]") and *Tt*FRS (orange; PDB "2IY5 [https://www.rcsb.org/structure/2IY5]") structures on *Pvc*FRS:BRD1389 complex (cornflower blue). **b** Opening of Loop 1 (residues 443–453) of *Pvc*FRS for BRD1389 binding to accommodate its methoxymethyl group and adoption of a possible open conformation. **c** Closing of Loop 2 (residues 507–515) in *Pvc*FRS:BRD1389 complex. **d** Open conformation of the Arg548 in *Pvc*FRS:BRD1389 complex accommodates its [6.2.0]-diazabicyclodecane core as compared to its orthologue residue in *Hsc*FRS (Arg463) and *Tt*FRS (Arg321) which in its closed conformation may clash with this ring.

1,6-diazabicyclo[6.2.0]dec-3-ene.[14] Please refer to the Supplementary Information for synthetic protocols and analytical data.

**Protein expression and purification**. Full-length *Pfc*FRS was purified according to the earlier published report[2]. Full-length *Pvc*FRS was also purified according to the same protocol. In brief, the genes encoding *Pvc*FRS alpha subunit (PVX_081300) and beta subunit (PVX_090880) were cloned into *E. coli* pETM11 and pETM20 plasmids respectively. Both plasmids were co-transformed into *E. coli* strain B834 and were induced overnight for overexpression with 0.5 mM isopropyl-β-D-thiogalactoside (IPTG) at 16 °C for 18 h. The *E. coli* cell lysate was first loaded onto a nickel–nitrilotriacetic (Ni–NTA) column (GE Healthcare) and the eluted fraction was further purified with Heparin chromatography (GE Healthcare) to a single band as indicated by SDS–polyacrylamide gel electrophoresis with Coomassie brilliant blue staining. The purified protein was found as a single peak with the elution volume consistent of a homogeneous *Pvc*FRS hetero-tetramer on the Superdex 200 analytical gel filtration column (GE Healthcare). The purified *Pvc*FRS

was concentrated to 25 mg ml⁻¹ and stored at −80 °C in 25 mM HEPES buffer, pH 7.5, 200 mM NaCl, 5 mM βME.

**Evaluation of parasite growth inhibition**. *Plasmodium falciparum* strain 3D7 (chloroquine–sensitive) and K1 (chloroquine-resistant) were obtained from Kitasato University and used for testing antimalarial activities in vitro. The cultivation of *P. falciparum* was conducted according to Trager's method with some modification[17]. Precisely, parasites were kept in culture flasks with RPMI1640 medium supplemented with 10% human plasma and 2% fresh human erythrocytes and incubated at 37 °C with the gas condition of 5% $CO_2$ and 5% $O_2$. The parasitemia (percentage of infected erythrocytes to total erythrocytes) were kept within 0.25–10%. Culture medium was replaced, and fresh erythrocytes were supplied every 2–3 days. Drug susceptibility test was conducted according to Desjardins's method[18] with some modification. The bicyclic azetidines and known antimalarial agents (chloroquine and artemisinin) were tested at the same time. Precisely, 199.5 μl of parasite cultures (2% hematocrit and 0.75–1% parasitemia) and 0.5 μl of compound solution serially diluted in DMSO were poured into every well in 96-well titer plates and final drug concentrations were set

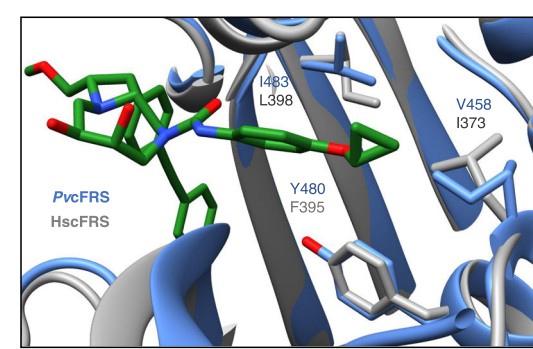

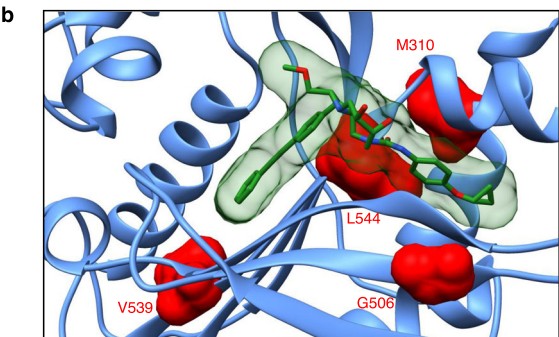

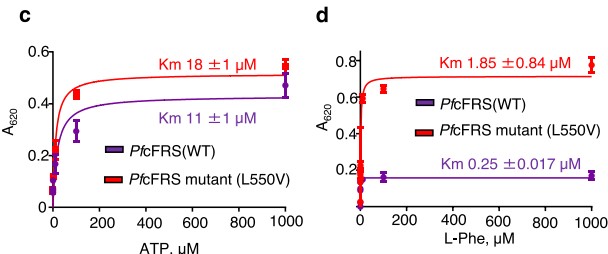

**Fig. 5 Resistance-conferring mutations are localized near the BRD1389 binding site. a** Close-up view of *Pvc*FRS (cornflower blue) with *Hsc*FRS (grey, PDB "3L4G [https://www.rcsb.org/structure/3l4g]") as superimposed structures. Non-conserved residues within 5 Å of BRD1389 are shown. **b** *Pvc*FRS structure with known *Pfc*FRS mutations are highlighted in red. All resistance mutations map within the confines of the active site region wherein the L550V mutant lies closest to the bound ligand. **c**, **d** Effect of L550V mutation on enzyme activity. Both enzymes show almost similar *Km* value for the ATP ($18 \pm 1\,\mu M$ for *Pfc*FRS wild type (WT) and $11 \pm 1\,\mu M$ for *Pfc*RS mutant (L550V)) whereas the *Km* value for L-Phe is increased ~10-fold for the mutant enzyme ($0.25 \pm 0.017\,\mu M$ for *Pfc*FRS (WT) and is $1.85 \pm 0.84\,\mu M$ for *Pfc*FRS mutant (L550V)). Data are shown as mean $\pm$ SD ($n = 3$ independent experiments). Source data are provided as a Source Data file.

within $0.001–1\,\mu g\,ml^{-1}$. The plates were kept at 37 °C with the gas condition of 5% $CO_2$ and 5% $O_2$ for 72 h and then parasite growth was quantified with Makler's method to detect plasmdoial lactate dehydrogenase[19] with some modification. Precisely, culture plates were kept in freezer overnight and then thawed at 37 °C to disrupt the erythrocytes and parasite cells. In the new 96-well titer plates, 100 μl of enzyme reaction solution (110 mM Li-lactate, 0.5 mM acetylpyridine-adenine dinucleotide, 50 mM Tris (pH 7.5), 10 mM EDTA, 50 mM KCl, and $15\,g\,l^{-1}$ PEG6000) and 20 μl of freeze-thaw culture were mixed in each well and then kept at room temperature for 30 min. The detection solution was prepared by mixing equal volume of $2\,mg\,ml^{-1}$ nitro blue tetrazolium and $0.1\,mg\,ml^{-1}$ phenazine ethosulfate and 20 μl of the solution was added to each well. After the incubation at room temperature in the dark for 90 min, absorbance at 660 nm was analyzed and $IC_{50}$s were calculated from dose response curve.

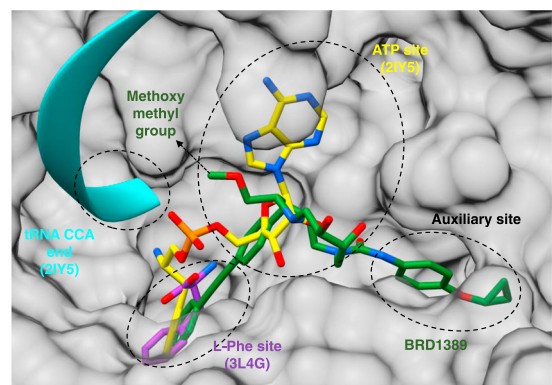

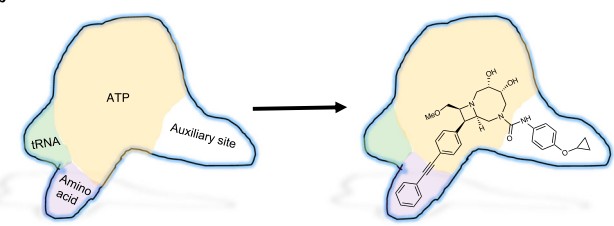

**Fig. 6 BRD1389 is a novel dual-site inhibitor of *Plasmodium* cFRS. a** Catalytic pocket in *Pvc*FRS:BRD1389 complex while it is superimposed on *Hsc*FRS-L-Phe (Purple; PDB "3L4G [https://www.rcsb.org/structure/3l4g]") and on *Tt*FRS-L-Phe-AMP-tRNA (Yellow; PDB "2IY5 [https://www.rcsb.org/structure/2IY5]"). BRD1389 occupies L-Phe and auxiliary sites and its methoxymethyl group (arrow; green) is proximal to ATP binding site. **b** Schematic map showing BRD1389 occupancy in the enzyme pockets.

**Enzyme inhibition assays**. These were done using malachite green assay as per earlier published reports[1,6]. Briefly, the reaction was performed for 100 μM ATP, 50 μM L-phenylalanine and 100 nM recombinant PheRS enzymes (*Pf, Pv, Hs, Pf* mutant (L550V)) in a buffer containing 30 mM HEPES (pH 7.5), 150 mM, NaCl, 30 mM KCl, 50 mM, $MgCl_2$, 1 mM DTT, and 2 U/ml *E. coli* inorganic pyrophosphatase (NEB) at 37 °C. Enzymatic reactions (50 μl total volume) were performed in clear, flat-bottomed, 96-well plates (Costar 96-well standard microplates). The reaction mixture was incubated for 2 h at 37 °C. The reaction was stopped by adding 12.5 μl of malachite green solution to the reaction mixture and levels of inorganic phosphate (Pi) were detected after incubation of 5 min at room temperature. Absorbance was the measured at 620 nm using a Spectramax M2 (Molecular Devices). Reactions without FRS enzyme were performed as background controls, values of which were subtracted from the reactions with enzyme. BRD1389 was added to our aminoacylation assay reaction buffer in varying concentrations ranging from 0.1 nM to 10,000 nM. The $IC_{50}$ value for the data is shown for three replicates as the mean $\pm$ SD.

**Surface plasmon resonance experiments**. These were carried out on a Biacore T200 instrument (GE Healthcare) at 25 °C. The binding experiments were performed in buffer 10 mM phosphate buffered saline (PBS), pH 7.4, containing 5% dimethyl sulfoxide (DMSO). The flow system was primed with the running buffer before the initiation of the experiment. Both *Pfc*FRS and *Hsc*FRS were immobilized to Sensor Chip CM5 by standard amine coupling chemistry using N-hydroxysuccinimide (NHS) and ethyl(dimethylaminopropyl) carbodiimide (EDC) to an immobilization level of approximately 1500 RU. The binding experiments were carried out in a single cycle kinetics mode. BRD1389 was serially diluted in running buffer and injected at a flow rate of $60\,\mu l\,min^{-1}$ across both surfaces for 60 s and dissociations were set up for 120 s. The data analysis was done using Biacore evaluation T200 Evaluation software (GE Healthcare) and after applying the solvent correction, the data was fitted into the 1:1 binding evaluation method to determine the equilibrium dissociation constants ($K_d$).

***Pfc*FRS mode of inhibition studies**. To establish the mode of inhibition of BRD1389, data sets (generated using the malachite Green assay platform as described in the above enzyme inhibition assays method section) were collected by varying both the inhibitor and substrate concentrations. Using Graph Pad Prism each data set was individually fitted to the Michaelis–Menten equation and the resulting Lineweaver-Burk plots were examined for diagnostic patterns of

competitive, mixed or uncompetitive inhibition. Data sets were then globally fitted to the appropriate model (with Eqs. 1 and 2 used for competitive and mixed inhibition respectively).

$$v = \frac{Vmax.[S]}{Km\left(1 + \frac{[I]}{Ki}\right) + [S]} \tag{1}$$

$$v = \frac{Vmax.[S]}{Km\left(1 + \frac{[I]}{Ki}\right) + Km\left(1 + \frac{[I]}{Ki}\right)[S]} \tag{2}$$

**Crystallization, data collection and structure determination**. The purified *Pfc*FRS and *Pvc*FRS proteins were used for crystallization by the hanging-drop vapour-diffusion method at 293 K using commercially available crystallization screens (Index, JCSG-plus, Morpheus, PACT premier, PGA, Crystal Screen, PEG/Ion and ProPlex; Hampton Research and Molecular Dimensions). Initial screening was performed in 96-well plates using a nano drop dispensing Mosquito robot (TTP Labtech). Three different drop ratios were used for the crystallization trials by mixing 75, 100, or 50 nl purified protein solution with 75, 50, or 100 nl reservoir solution, respectively (i.e., 1:1, 2:1, and 1:2 drop ratios). Each of the drops was equilibrated against 100 ml of the corresponding reservoir solution. Before crystallization, *Pvc*FRS was diluted to 12 mg ml$^{-1}$ with 3 mM BRD1389, 5 mM MgCl$_2$, and 4 mM βME, and then incubated on ice for 30 min.

Diffraction quality *Pvc*FRS-BRD1389 crystals were obtained in PGA screen F4 [0.1 M sodium cacodylate (pH 6.5), 3% w/v poly-γ-glutamic acid (Na$^+$ form, low molecular weight), 3% w/v PEG20000, 0.1 M ammonium sulphate, 0.3 M sodium formate]. The crystals were mounted in nylon loops (Hampton Research) or litho loops (Molecular Dimensions) after being soaked for 10–30 s in a cryoprotectant containing the corresponding crystallization mother liquor with 20%(v/v) glycerol. The crystals were subsequently flash-cooled in liquid nitrogen. X-ray diffraction data set were collected on beamline I24 at Diamond Light Source (DLS), United Kingdom at a wavelength of 0.9688 A˚. The data were processed by the xia2 auto-processing pipeline[20] using DIALS[21] for integration. The initial model for *Pv*FRS-BRD1389 was determined by the molecular-replacement (MR) method using Phaser[22] with *Hs*FRS (PDB entry "3L4G_OP [https://www.rcsb.org/structure/3l4g]") as the template. It was then subjected to AutoBuild[23] that provided a partial model with R$_{work}$/R$_{free}$ of 31/41% for ~800 residues in 29 fragments. Subsequently, the model was manually built and completed by iterative cycles of building using COOT[24] and refinement using Refmac[25]. After each cycle of manual building and refinement, the models were inspected and manually adjusted to correspond to the 2F$_o$-F$_c$ and F$_o$-F$_c$ electron density maps. During refinement, the ligands BRD1389 and Mg$^{2+}$ ion were added based on positive peaks in difference Fourier maps and the model was subjected simulated annealing refinement using *phenix.refine* in Phenix[26]. The final model was refined to 3.0 Å resolution with R$_{work}$/R$_{free}$ of 21.4/28.8%. The stereo-chemical quality of the model was analysed using MolProbity[27] and the model has good geometry quality and all residues are in favoured/allowed (92/8%) regions of the Ramachandran plot. We additionally carried out crystal packing analysis of *Pv*FRS-BRD1389 complex using COOT[24]. Statistics of data collection and structure refinement are given in Supplementary Table 1. The atomic coordinates and structural factors have been deposited into Protein Data Bank with accession code "7BY6 [https://www.rcsb.org/structure/7BY6]".

**Kinetic Parameter Determination**. Michaelis constant, *Km* for the ATP and L-Phe substrates for *Pfc*FRS wild type (WT) and *Pfc*FRS mutant (L550V) was determined using the malachite green assay[6] in a buffer containing 30 mM HEPES (pH 7.5), 150 mM, NaCl, 30 mM KCl, 50 mM, MgCl$_2$, 1 mM DTT, and 2 U/ml *E. coli* inorganic pyrophosphatase (NEB). ATP *Km* was determined using saturating concentration of L-Phe (1000 μM) and varying the concentration of ATP from 1000 to 0.01 μM. Similarly, the *Km* for L-Phe was determined using saturating concentration of ATP (500 μM) and varying the concentration of L-Phe from 1000 to 0.0001 μM. In both cases the reaction was carried out for 2 h at 37 °C. Data were fitted to Michaelis–Menten equations using Prism graph 6.0 software.

**Sequence analysis and structure presentation**. Protein sequences were aligned using the program Cluster Omega[28] by using the default settings. All structural superimpositions and preparation of figures was conducted using Chimera[29] and Pymol[30].

**Reporting summary**. Further information on research design is available in the Nature Research Reporting Summary linked to this article.

## Data availability
The authors confirm that all relevant biochemical and synthesis data supporting the findings are included in the paper and its supplementary file. Structural data have been deposited into Protein Data Bank with accession code "7BY6 [https://www.rcsb.org/structure/7BY6]". Public datasets ("3L4G [https://www.rcsb.org/structure/3l4g]", "2IY5

[https://www.rcsb.org/structure/2IY5]") were used in this study. Source data for figures [Table 1, Figs. 1b–d, 5c, d and Supplementary Figs. 3b, 4b] are provided with this paper.

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

## Acknowledgements

This work was supported by the Global Health Innovative Technology Fund (GHIT Fund, grant G2016-219 to S.L.S.). AS laboratory is supported by Medicines for Malaria Venture (MMV), DBT (PR32713) and DST via JC Bose fellowship. The authors thank R. Philbin (Broad Institute) for assessment of compound purity, T. Horii (Eisai Co., Ltd.) for evaluation of in vitro antiparasitic potency, S.-H. Bae and J. Chen (Eisai, Inc.) for helpful comments on the manuscript, C. Hon (Broad Institute) for project management, and F. Gusovsky (Eisai, Inc.) for project leadership.

## Author contributions

A.S. designed and managed the study. M.S., N.M., M.Y., and A.S. purified the protein, crystallized, and solved the structure. M.S. performed all biochemical assays. K.H. helped with data collection and S.P. helped with biochemical analysis. B.Me., E.C., A.G., B.Mi., F. G.F., and S.L.S. designed and synthesized BRD1389. We wrote the manuscript.

## Competing interests

S.L.S. is a shareholder and serves on the Board of Directors of Jnana Therapeutics; is a shareholder of Forma Therapeutics and Decibel Therapeutics; is a shareholder and advises Kojin Therapeutics, Kisbee Therapeutics and Eikonizo Therapeutics; serves on the Scientific Advisory Boards of Eisai Co., Ltd., Ono Pharma Foundation, Exo Therapeutics, Biogen, Inc. and F-Prime Capital Partners; and the Board of Advisers of the Genomics Institute of the Novartis Research Foundation; and is a Novartis Faculty Scholar. The Broad Institute and Harvard have filed patent applications relating to work described in this manuscript, including "Compounds and Methods for Treating Parasitic Diseases" (PCT/US18/23270, filed on March 20, 2018 and published as WO18/175385), and "Apicomplexan Parasite Inhibition" (unpublished).
