## [Peer Review File · Nature Communications]

Reviewer comments, first round -

Reviewer #1 (Remarks to the Author):

In the manuscript entitled "Structural basis of malaria parasite phenylalanine tRNA-synthetase inhibition by bicyclic azetidines", the authors describe the mechanism of inhibition of a bicyclic azetidine toward Plasmodium vivax cytosolic phenylalanine-t-RNA-synthetase (cFRS) and explore the structural aspects involved in the ligand-enzyme molecular recognition. The compound described in this study is a derivative of a previously reported series that proved to be highly active against different Plasmodium species, and effective against a P. falciparum murine model of malaria. Additionally, structural features that lead to selectivity over human cFRS and resistance to antimalarial agents are explored.

The manuscript is the first to describe the structure of Plasmodium vivax phenylalanine-tRNA-synthetase in complex with an inhibitor, although with a resolution (3 Å) well above the desirable range. The authors additionally show that the binding mode of the compound provides a novel path to the design of selective molecules with respect to the human enzyme.

The manuscript can be improved in several aspects. First, the validation of cFRS as a drug target should be briefly discussed. Is there any controversy in the literature regarding targeting aminoacyl RNA synthetases in malaria drug discovery? What has been done to validate genetically and chemically the enzyme as a molecular target? These aspects should be explored.

Does cFRS meet druggability criteria for further drug discovery efforts? This should be clearly stated.

The authors suggest that loop displacements occur because of ligand binding. However, there is the possibility of random loop fluctuations considering that these highly flexible elements are usually located at protein surfaces, and considering the limitations of x-ray crystallography. This point should be clarified.

The sentence "A recent discovery of chemical series comprising a bicyclic azetidine scaffold that exhibits multistage antimalarial activity, and can achieve single-dose cures in a mouse model of malaria" must be rephrased.

Were control compounds used in the enzymatic and binding assays?

What was the PfcFRS solution concentration used for the crystallization experiments? Why PfcFRS structure has not been solved?

Further details on sequence alignment procedures should be added to the manuscript.

Considering that the study focuses on only one ligand, why HPLC runs for compound purity checking have not been carried out?

Apart from the above highlighted comments, the manuscript brings new and relevant data for further efforts regarding malaria drug discovery and development.

Reviewer #2 (Remarks to the Author):

The manuscript entitled "Structural basis of malaria parasite phenylalanine tRNA-synthetase inhibition by bicyclic azetidines" describes the structural and biochemical evidence that bicyclic azetidines are competitive inhibitors of L-Phe, required for the enzyme-catalyzed aminoacylation reaction that underpins protein synthesis in the parasite. Moreover, the manuscript indicates residue variations between the malaria parasite tRNA-synthetase and the human homolog that underpin the highly selective enzyme inhibition and parasite killing by bicyclic azetidines.

The authors present appealing data from a combination of biochemical and crystallographic studies that revealed the molecular underpinnings of Plasmodium tRNA-synthetase inhibition by bicyclic azetidines. The use of Plasmodium vivax phenylalanine tRNA-synthetase as a surrogate for the Plasmodium falciparum homolog is well-justified. The crystallographic structure of the phenylalanine tRNA-synthetase in complex with the potent bicyclic azetidine representative

inhibitor and the mutagenic and biochemical studies conducted lays a foundation for designing the next generation of phenylalanine tRNA-synthetase inhibitors as antimalarial drug candidates.

The collection and refinement statistics are satisfactory. The phenylalanine tRNA-synthetase complex structure is of modest quality (refined to 3.0 angstrom). The number of amino acid residues analyzed for Ramachandran outliers, rotameric score, model fitting is no more than 50% of the total amino acids for alpha subunit, which skews the statistical analyses. However, the observed electronic density within the enzyme catalytic site of alpha subunit was of enough quality to unambiguously fit the ligand to experimental data, thereby indicating the binding mode of the bicyclic azetidine derivative to phenylalanine tRNA-synthetase. Based on that, the inhibitor occupies both the L-Phe site and an auxiliary pocket within phenylalanine tRNA-synthetase, which represents a novel dual-site malaria parasite tRNA-synthetase inhibitor.

The technical quality of the research reported is valid and appropriate. The degree of novelty and originality is high, and the conclusions are adequately supported by the data presented. Taken together, the data presented provides an important advance in the investigation of phenylalanine tRNA-synthetase as an attractive and validated molecular target for the discovery of new antimalarial drugs and, potentially, other diseases caused by apicomplexans, including toxoplasmosis and cryptosporidiosis. Therefore, the work will be interesting for the antimalarial community as new antimalarial targets and inhibitors are needed to be clinically validated. So, I am happy to suggest it for acceptance. Nevertheless, there are some specific points that the authors should modify or reconsider previously to the publication of the work.

Please find below my comments:

- Given the higher potency of BRD1389 against the whole parasite (Dd2 growth inhibitory activity of < 1 nM) related to the evaluated potency against the isolated Pf phenylalanine tRNA-synthetase (IC₅₀ = 12nM), the authors should comment whether a secondary mechanism of action is expected for the bicyclic azetidines.

Specific comments:

- Page 11. The authors should add the standard deviation for data in Table 1, Extended Data 1C, and 1D.

Rafael V. C. Guido

Reviewer #3 (Remarks to the Author):

In 2016, bicyclic azetidines targeting phenylalanyl-tRNA synthetase (cFRS) were described as attractive new antimalarials. The current study describes the X-ray crystal structure of Plasmodium vivax cFRS bound to one of the compound series (BRD1389) and identifies key regions of the new drug target important for inhibitor binding. The inhibitory kinetics of the compound are characterised (activity against parasites is previously reported), and structural basis for selectivity against human homologs investigated.

The data reported is new and will be of interest to the future design of antimalarial agents targeting cFRS, as well as potentially other anti-parasitic drug targets. The new structure identifies the binding site of these compounds and articulates the mechanism of inhibition. Key differences that provide selectivity between the human and plasmodium enzymes are identified, and residues contributing to compound binding and/or stabilisation are also identified.

The structure of the vivax enzyme was solved via molecular replacement using the human homolog coordinates (3L4G.pdb). The human structure has a similar resolution (3.3 Ang to 3.0 as described for PvcFRS) and overall structure. Key differences in quaternary structure are described.

The data obtained has a moderate resolution (3.0 Ang reported), and a structure with one heterodimer in the au, however, only ~ 50% of one monomer is modelled (Chain A = 296 residues

or approx 50% of the chain, Chain B = 602 residues). The text states that there was no density observed for the N-terminal DNA binding domain of chain A. Overall refinement statistics show an Rfree of 28.2, which is a little surprising given the resolution is limited to 3.0 Ang and a significant proportion of asymmetric unit remains unmodelled (~25% of au has no modelled density). Investigation of the MR probe, 3L4G, shows similar refinement statistics. Molprobity analysis of 3L4G identifies significant causes for concern with regard to the quality of this structure (rated only in the 47th percentile of structures at this resolution). Therefore, I fear that 3L4G may be over-refined for its data resolution and this bias may be present in the new vivax structure. Reading of the methods provided, does not suggest that any simulated annealing was performed post Phaser and prior to the initial build of this structure. How was the MR model of 3L4G prepared for use as the probe? Figure 3d lists a simulated anneal omit map - did the authors inspect other areas of the structure (beyond the compound binding region) to inspect if this resulted in any changes? Whilst the structure presented here has excellent quality scores, the R/Rfree indicate possible over-refinement (particularly with such disorder arising from absent domains).

A simulated anneal omit map identifies the binding site of BRD1389 which has patchy and non-connected electron density at 1 sigma. A best fit is shown in Figure 3d but is not unambiguous with regard to orientation of key elements of the compound. Taken together with overlays of other structures bound to their natural substrates (Fig 3c) and mutational resistance data, it shows the likely pose and orientation of the compound.

Figure 4 discusses "highly noticeable" loop openings and closing mediated by two key residues - Arg548 and His451. From the figures provided, the movement of Arg548 is clear but the histidines are less obvious and certainly do not appear to represent "major loop distortions". All 3 structures that are superimposed are of similar resolution and therefore, such loop areas but be subject to limitations of resolution as well as normal loop fluctuations within static structures. After reading the section and inspecting the figure 4, I am still unsure what the take home message is and if it is really as significant as the language used in parts would indicate. I wonder if this very long paragraph describing this section could be broken up and clarified to the reader, being wary of over-interpreting the structural data that is available.

Minor points

Subheadings would be of great benefit to this article. Not sure if they are allowed by the journal but currently, there is little sign-posting of key findings making it difficult to follow (for what is a long article with many figure panels).

Figure 1i and j. Figure legend needs additional information. What is the orange and green in this panel? Is it possible to orient the purple and grey part of the structure in the same orientation as panels e, f, g & h as this would help the reader see the key points being made (which I think is the differences in the domain swaps in the formation of the tetramer?

The SPR shown in Extended Data Figure 1 shows the comparison of PfcFRS vs HscFRS, with what appears to be a much slower off rates for Pf than Hs, and does not appear to return to baseline. Is ligand binding fully reversible? Or is there some evidence of long duration association or other interactions?

Please find below point-wise responses (in **black**) to reviewers queries/comments (in **blue**):

Reviewers' comments:

Reviewer #1 (Remarks to the Author):

In the manuscript entitled "Structural basis of malaria parasite phenylalanine tRNA synthetase inhibition by bicyclic azetidines", the authors describe the mechanism of inhibition of a bicyclic azetidine toward *Plasmodium vivax* cytosolic phenylalanine-tRNA-synthetase (cFRS) and explore the structural aspects involved in the ligand-enzyme molecular recognition. The compound described in this study is a derivative of a previously reported series that proved to be highly active against different *Plasmodium* species, and effective against a *P. falciparum* murine model of malaria. Additionally, structural features that lead to selectivity over human cFRS and resistance to antimalarial agents are explored. The manuscript is the first to describe the structure of *Plasmodium vivax* phenylalanine-tRNA-synthetase in complex with an inhibitor, although with a resolution (3 Å) well above the desirable range. The authors additionally show that the binding mode of the compound provides a novel path to the design of selective molecules with respect to the human enzyme. The manuscript can be improved in several aspects.

COMMENTS and QUESTIONS

1. First, the validation of cFRS as a drug target should be briefly discussed.

In the revised paper we have added a line about cFRS as a validated drug target and also added the reference for the same. The added line reads – The *Pfc*fRS was validated both genetically and biochemically as a drug target for the bicyclic azetidine series of molecules as shown in our previous study. This has been added on page 3 of main text.

Ref: Kato, N. et al. Diversity-oriented synthesis yields novel multistage antimalarial inhibitors. *Nature* 538, 344-349 (2016).

2. Is there any controversy in the literature regarding targeting aminoacyl RNA synthetases in malaria drug discovery?

On the contrary, there are numerous laboratories worldwide that are now targeting aminoacyl RNA synthetases of malaria parasites for drug discovery. Even for this work we are aware of very stiff competition, and we are aware of advanced work on other targets within the aminoacyl tRNA synthetase family. Unrelatedly, tavaborole is an antifungal agent that inhibits leucyl-tRNA synthetase and is now a US FDA approved drug for treatment of onychomycosis. So, there is sufficient enthusiasm for drugging the malaria parasite encoded aminoacyl RNA synthetases. Please see few references that support targeting of aminoacyl RNA synthetases in malaria drug discovery:

1. Ref: Baragaña, B. et al. Lysyl-tRNA synthetase as a drug target in malaria and cryptosporidiosis. *Proc. Natl. Acad. Sci. USA*. **116**, 7015-7020 (2019).
2. Ref: Manickam, Y. et al. Drug targeting of one or more aminoacyl-tRNA synthetase in the malaria parasite *Plasmodium falciparum*. *Drug Discov. Today* **23**, 1233-1240 (2018).

3. Ref: Kato, N. et al. Diversity-oriented synthesis yields novel multistage antimalarial inhibitors. *Nature* 538, 344-349 (2016).
4. Ref: Herman, J. D. et al. The cytoplasmic prolyl-tRNA synthetase of the malaria parasite is a dual-stage target of febrifugine and its analogs. *Sci Transl Med.* 7, 288ra77 (2015)
5. Ref: Jain, V. et al. Structure of Prolyl-tRNA Synthetase-Halofuginone Complex Provides Basis for Development of Drugs against Malaria and Toxoplasmosis. *Structure* 23, 819-829 (2015).
6. Ref: Khan, S. et al. Structural basis of malaria parasite lysyl-tRNA synthetase inhibition by cladosporin. *J. Struct. Funct. Genomics* 15, 63-71 (2014).
7. Ref: Hoepfner, D. et al. Selective and specific inhibition of the plasmodium falciparum lysyl-tRNA synthetase by the fungal secondary metabolite cladosporin. *Cell Host Microbe.* 11, 654-663 (2012).

3. What has been done to validate genetically and chemically the enzyme as a molecular target? These aspects should be explored

Phenotypically, biochemically and via selection of drug-resistant mutations the *PfcFRS* has been validated as a drug target. As a matter of fact, this protein target has been awarded two GHIT grants (one very recently in 2020) for drug development against malaria parasites.

Ref: Kato, N. et al. Diversity-oriented synthesis yields novel multistage antimalarial inhibitors. *Nature* 538, 344-349 (2016).

4. Does cFRS meet druggability criteria for further drug discovery efforts? This should be clearly stated.

Indeed so. The cFRS meets the druggability criteria for further drug discovery as 1) bicyclic azetidine drugs inhibit plasmodium cFRS activity with high specificity (Kato et al 2016), 2) its 3-D structure is available now (via this paper) and this will set up further structure-based drug design, 3) cFRS is inhibited by bicyclic azetidine drugs with high potency (this and previous studies), 4) drug resistance mutations generated in parasites show the gene for cFRS as the target, and 5) this protein target has been awarded two GHIT grants (one in 2020) for drug development against malaria parasites. The text has been revised to include reviewer's suggestion on page 3 of main text.

5. The authors suggest that loop displacements occur because of ligand binding. However, there is the possibility of random loop fluctuations considering that these highly flexible elements are usually located at protein surfaces, and considering the limitations of x-ray crystallography. This point should be clarified.

We understand the reviewers concern. However presence of both loops (Loop1 and Loop2) in proximity of BRD1389 binding site together with structural comparison along the loop regions in *PvcFRS*-BRD1389 versus *HsFRS*-Lphe (PDB-3LAG) (Figure 4 is revised, Supplementary Figure 6b, 6c) suggests possible loop opening and closing. The electron density figures have been added as Supplementary Figure 6 and the text has been revised to highlight these points on page 7. The Figure 4 has been revised to include reviewers suggestion.

6. The sentence "A recent discovery of chemical series comprising a bicyclic azetidine

scaffold that exhibits multistage antimalarial activity, and can achieve single-dose cures in a mouse model of malaria" must be rephrased.

According to reviewer's suggestion, the above sentence is now rephrased as:

Recently, a series of small molecules based on a bicyclic azetidine core has been discovered that exhibits multistage antimalarial activity and can achieve single-dose cures in a mouse model of malaria. This has been changed on page 3.

Ref: Kato, N. et al. Diversity-oriented synthesis yields novel multistage antimalarial inhibitors. *Nature* 538, 344-349 (2016).

7. Were control compounds used in the enzymatic and binding assays?

Yes, we used appropriate controls in all assays. For enzymatic assay, recombinant maltose-binding protein (MBP) or no protein was used as controls (as published by *Sharma et al, see full reference below*). For binding studies ATP + L-Phe (1 mM each) were used as controls in place of the BRD drugs.

Ref: Sharma, A. & Sharma, A. Plasmodium falciparum mitochondria import tRNAs along with an active phenylalanyl-tRNA synthetase. *Biochem. J.* **465**, 459–469 (2015).

8. What was the *PfcFRS* solution concentration used for the crystallization experiments? Why *PfcFRS* structure has not been solved?

We have used $\sim 5 \text{ mg ml}^{-1}$ concentration for *PfcFRS* crystallization. We also have crystals of *PfcFRS* but the resolution is so far very limiting.

9. Further details on sequence alignment procedures should be added to the manuscript.

The details about the procedure and corresponding reference has been added in the revised text on page 29.

10. Considering that the study focuses on only one ligand, why HPLC runs for compound purity checking have not been carried out?

Indeed HPLC was carried out to check the purity of the drug used. We have added the HPLC details in our supplementary methods section and have added its profile in the Supplementary Figure 2.

Apart from the above highlighted comments, the manuscript brings new and relevant data for further efforts regarding malaria drug discovery and development.

We thank the reviewer for this positive response.

Reviewer #2 (Remarks to the Author):

The manuscript entitled “Structural basis of malaria parasite phenylalanine tRNA-synthetase inhibition by bicyclic azetidines” describes the structural and biochemical evidence that bicyclic azetidines are competitive inhibitors of L-Phe, required for the enzyme-catalyzed aminoacylation reaction that underpins protein synthesis in the parasite. Moreover, the manuscript indicates residue variations between the malaria parasite tRNA-synthetase and the human homolog that underpin the highly selective enzyme inhibition

and parasite killing by bicyclic azetidines. The authors present appealing data from a combination of biochemical and crystallographic studies that revealed the molecular underpinnings of Plasmodium tRNA-synthetase inhibition by bicyclic azetidines. The use of Plasmodium vivax phenylalanine tRNA-synthetase as a surrogate for the Plasmodium falciparum homolog is well-justified. The crystallographic structure of the phenylalanine tRNA-synthetase in complex with the potent bicyclic azetidine representative inhibitor and the mutagenic and biochemical studies conducted lays a foundation for designing the next generation of phenylalanine tRNA-synthetase inhibitors as antimalarial drug candidates. The collection and refinement statistics are satisfactory. The phenylalanine tRNA-synthetase complex structure is of modest quality (refined to 3.0 angstrom). The number of amino acid residues analyzed for Ramachandran outliers, rotameric score, model fitting is no more than 50% of the total amino acids for alpha subunit, which skews the statistical analyses. However, the observed electronic density within the enzyme catalytic site of alpha subunit was of enough quality to unambiguously fit the ligand to experimental data, thereby indicating the binding mode of the bicyclic azetidine derivative to phenylalanine tRNA-synthetase. Based on that, the inhibitor occupies both the L-Phe site and an auxiliary pocket within phenylalanine tRNA-synthetase, which represents a novel dual-site malaria parasite tRNA-synthetase inhibitor. The technical quality of the research reported is valid and appropriate. The degree of novelty and originality is high, and the conclusions are adequately supported by the data presented. Taken together, the data presented provides an important advance in the investigation of phenylalanine tRNA-synthetase as an attractive and validated molecular target for the discovery of new antimalarial drugs and, potentially, other diseases caused by apicomplexans, including toxoplasmosis and cryptosporidiosis. Therefore, the work will be interesting for the antimalarial community as new antimalarial targets and inhibitors are needed to be clinically validated. So, I am happy to suggest it for

acceptance. Nevertheless, there are some specific points that the authors should modify or reconsider previously to the publication of the work.

We thank the reviewer for this positive response. Indeed, our work has implications for toxoplasmosis and cryptosporidiosis. Moreover, the structure determination procedure has now been elaborated further and revised text has been added as shown below in the methods section on page 29.

We have additionally carried out crystal packing analysis of *Pv*FRS-BRD1389 complex and found no room for the missing N-terminal domain of the alpha subunit (residues 1-270). A crystal packing figure (Supplementary Figure 4a) has been added in the supplementary section. Further, we carried out SDS-PAGE analysis of *Pvc*FRS protein to determine the actual size of both the subunits of the protein in the crystal lattice and we found two bands at ~71.4 kDa and ~40 kDa (Supplementary Figure 4b). These two band sizes corresponds to the full-length beta subunit (618 residues, ~71.4 kDa) and a shortened alpha subunit, catalytic domain (residues 271-569, ~40 kDa) respectively. Thus, both crystal packing analysis and SDS-PAGE clearly suggest that the crystallized size of the alpha subunit is ~40 kDa (Supplementary Figure 4b). Hence, there are no missing 270 residues of the alpha subunit to account for in the crystal structure analysis. This information has been added in the revised main text on page 5-6 and both crystal packing and SDS-PAGE gel pictures have been added in the supplementary section (Supplementary Figure 4). The atomic coordinates of the PDB entry 7BY6 have been revised and the sequence information of the alpha subunit has been updated. The updated validation report is attached.

COMMENTS and QUESTIONS

1. Given the higher potency of BRD1389 against the whole parasite (Dd2 growth inhibitory activity of < 1 nM) related to the evaluated potency against the isolated *Pf* phenylalanine tRNA-synthetase (IC₅₀ = 12nM), the authors should comment whether a secondary mechanism of action is expected for the bicyclic azetidines.

Based on selection of parasite mutants that were partially resistant to cFRS (Kato *et al*, 2016), it seems that so far the only validated target of bicyclic azetidines is cFRS. We do not wish to speculate more on this score however it is noteworthy that:

- (1) The cFRS has been validated as a drug target both genetically and biochemically for bicyclic azetidine molecules as shown in our previous published work (Kato *et al*, 2016).
- (2) Also, we have shown very good correlation using bicyclic azetidine drugs for EC₅₀ and IC₅₀ values in our pervious published work (Kato *et al*, 2016).

Ref: Kato, N. et al. Diversity-oriented synthesis yields novel multistage antimalarial inhibitors. *Nature* 538, 344-349 (2016).

2. Page 11. The authors should add the standard deviation for data in Table 1, Extended Data 1C, and 1D.

Where possible standard deviations have been added and the figures have been revised . We have also included additional growth inhibition data (from different *Plasmodium* strains).

Reviewer #3 (Remarks to the Author):

In 2016, bicyclic azetidine compounds targeting phenylalanyl-tRNA synthetase (cFRS) were described as attractive new antimalarials. The current study describes the X-ray crystal structure of *Plasmodium vivax* cFRS bound to one of the compound series (BRD1389) and identifies key regions of the new drug target important for inhibitor binding. The inhibitory kinetics of the compound are characterised (activity against parasites is previously reported), and structural basis for selectivity against human homologs investigated. The data reported is new and will be of interest to the future design of antimalarial agents targeting cFRS, as well as potentially other anti-parasitic drug targets. The new structure identifies the binding site of these compounds and articulates the mechanism of inhibition. Key differences that provide selectivity between the human and plasmodium enzymes are identified, and residues contributing to compound binding and/or stabilisation are also identified. The structure of the vivax enzyme was solved via molecular replacement using the human homolog coordinates (3L4G.pdb). The human structure has a similar resolution (3.3 Ang to 3.0 as described for *PvcFRS*) and overall structure. Key differences in quaternary structure are described.

COMMENTS and QUESTIONS

1. The data obtained has a moderate resolution (3.0 Ang reported), and a structure with one heterodimer in the au, however, only ~ 50% of one monomer is modelled (Chain A = 296 residues or approx 50% of the chain, Chain B = 602 residues). The text states that there was no density observed for the N-terminal DNA binding domain of chain A. Overall refinement statistics show an Rfree of 28.2, which is a little surprising given the resolution

is limited to 3.0 Ang and a significant proportion of asymmetric unit remains unmodelled (~25% of au has no modelled density).

We have additionally carried out crystal packing analysis of *Pv*FRS-BRD1389 complex and found no room for the missing N-terminal domain of the alpha subunit (residues 1-270). A crystal packing figure (Supplementary Figure 4a) has been added in the extended/supplementary section. Further, we carried out SDS-PAGE analysis of *Pvc*FRS protein to determine the actual size of both the subunits of the protein in the crystal lattice and we found two bands at ~71.4 kDa and ~40 kDa (Supplementary Figure 4b). These two band sizes corresponds to the full-length beta subunit (618 residues, ~71.4 kDa) and a shortened alpha subunit, catalytic domain (residues 271-569, ~40 kDa) respectively. Thus, both crystal packing analysis and SDS-PAGE clearly suggest that the crystallized size of the alpha subunit is ~40 kDa (Supplementary Figure 4). Hence, there are no missing 270 residues of the alpha subunit to account for in the crystal structure analysis.

This proteolysis information has been added in the revised main text on page 5-6 and both crystal packing and SDS-PAGE gel pictures have been added in the supplementary section (Supplementary Figure 4). The atomic coordinates of the PDB entry 7BY6 have been revised and the sequence information of the alpha subunit has been updated. The updated validation report is attached.

2. Investigation of the MR probe, 3L4G, shows similar refinement statistics.

Molprobrity analysis of 3L4G identifies significant causes for concern with regard to the quality of this structure (rated only in the 47th percentile of structures at this resolution). Therefore, I fear that 3L4G may be over-refined for its data resolution and this bias may be present in the new vivax structure.

We have addressed this concern. Please note that the deposited *HsFRS* (PDB ID: 3L4G) structure has 16 chains (A to P, eight α subunit and eight β subunit) and has 4 tetrameric assembly built of two $\alpha\beta$ heterodimers per asymmetric unit (ASU). In it, the N-terminal stretch of 1-180 residues fold into DNA binding domain. Amongst the eight α subunit in the ASU for *HsFRS*, only the C chain has this DNA binding fold portion. Therefore, we took two hetero dimers with or without DNA binding fold (CD and OP chains) of the *HsFRS* (PDB ID: 3L4G) as a template for molecular replacement solution of *PvFRS*-BRD1389 complex. The LLG and TFZ score for the obtained solution using CD template is 435 and 22 respectively, while the corresponding Matthews coefficient (as published by *Matthews in Molecular Biology, see full reference below*) along with solvent content are $2.24 \text{ \AA}^3 \text{ Da}^{-1}$ and 45% respectively. On the other hand, the LLG and TFZ score for the obtained solution using OP template is 474 and 23 respectively and the corresponding Matthews coefficient (as published by *Matthews in Molecular Biology, see full reference below*) and solvent content are $2.93 \text{ \AA}^3 \text{ Da}^{-1}$ and 58% respectively.

After the MR solution, the model was subjected to automatic model building module, AutoBuild in PHENIX that provided a partial model with ~ 800 residues in 29 fragments. Subsequently, the model was manually built and completed by iterative cycles of building using COOT) and refinement using REFMAC. After each cycles of manual building and refinement, the models were inspected and manually adjusted to correspond to the $2F_o-F_c$ and F_o-F_c electron density maps. During refinement, the ligand BRD1389 and Mg^{2+} ion were added based on positive peaks in difference Fourier maps and the model was subjected simulated annealing refinement using *phenix.refine* in PHENIX.

Furthermore, our present analysis of the stored enzyme and obtained crystals using SDS-PAGE gels clearly suggests that the N-terminal stretch of 1-270 residues has been cleaved during crystallization process (Supplementary Figure 4b). In addition, the crystal packing analysis also revealed that there is no room for the N-terminal domain of α subunit (Supplementary Figure 4b). Therefore, there is no possibility for the over refinement of the model. The representative regions of the final model with overlaid different electron density maps (Supplementary Figure 5 and 6) are now included in the revised supplementary section. These statements have been added in the revised text on page 29 and supporting figures were added in the revised supplementary section. The revised statistics of data collection and structure refinement are updated in Supplementary Table 1.

Ref: Matthews, B. W. Solvent content of protein crystals. *J Mol Biol.* **33**, 491-497 (1968).

3. Reading of the methods provided, does not suggest that any simulated annealing was performed post Phaser and prior to the initial build of this structure.

We apologize for not including these details in the original submission though we indeed used simulate annealing throughout the refinement process. An elaborated structure determination procedure now has been added in the revised text in the methods section on page 29. The atomic coordinates of the PDB entry 7BY6 have been revised and the sequence information of the α subunit has been updated. The updated validation report is attached here. The revised statistics of data collection and structure refinement are updated in Supplementary Table 1.

4. How was the MR model of 3L4G prepared for use as the probe?

The deposited *Hs*FRS (PDB ID: 3L4G) structure has 16 chains (A to P, eight α subunit and eight β subunit) and has 4 tetrameric assembly built of two $\alpha\beta$ heterodimers per asymmetric unit (ASU). In it, the N-terminal stretch of 1-180 residues fold into DNA binding domain. Amongst the eight α subunit in the ASU for *Hs*FRS, only the C chain has this DNA binding fold portion. Therefore, we took two hetero dimers with or without DNA binding fold (CD and OP chains) of the *Hs*FRS (PDB ID: 3L4G) as a template for molecular replacement solution of *Pvc*FRS-BRD1389 complex. The LLG and TFZ score for the obtained solution using CD template is 435 and 22 respectively, while the corresponding Matthews coefficient (as published by *Matthews in Molecular Biology, see full reference below*) along with solvent content are 2.24 $\text{\AA}^3 \text{Da}^{-1}$ and 45% respectively. On the other hand, the LLG and TFZ score for the obtained solution using OP template is 474 and 23 respectively and the corresponding Matthews coefficient (as published by *Matthews in Molecular Biology, see full reference below*) and solvent content are 2.93 $\text{\AA}^3 \text{Da}^{-1}$ and 58% respectively.

5. Figure 3d lists a simulated anneal omit map - did the authors inspect other areas of the structure (beyond the compound binding region) to inspect if this resulted in any changes?

In the revised paper, we have shown the overlapping composite simulated annealed omit (SA-omit) and the final $2F_o - F_c$ map for regions beyond active site – for example the connecting loop of the $\beta 1$ and $\beta 2$ sub-domains (Supplementary Figure 6a). We show the same for unique insertions in the $\beta 2$ sub-domains of the beta subunit of *Pvc*FRS-BRD1389 complex (Supplementary Figure 5). The electron density maps of the representative regions (Supplementary Figure 5) have been added in the revised supplementary section,

while the map for bound BRD1389 of the final model has been updated in Figure 2d. This has been revised in the main text of the manuscript on page 6.

6. Whilst the structure presented here has excellent quality scores, the R/Rfree indicate possible over-refinement (particularly with such disorder arising from absent domains).

Our new analysis shows that there is no over-refinement as the 1-270 residues of the alpha subunit are not present in the crystal. We below detail the full procedure for structure solution and refinement. There is no scope for over-refinement as simulated annealing was used regularly during the model building process. We have added representative maps to show the quality of the final model and all statistics are as per acceptable standards of PDB.

The initial model of the *Pv*FRS-BRD1389 complex was determined by molecular replacement (MR) method using the $\alpha\beta$ hetero dimer (OP chains) of the *Hs*FRS (PDB ID: 3L4G) as a template. The PHASER (as published by McCoy et al in crystallography, see full reference below) program in PHENIX (as published by Adams et al in Acta crystallography, see full reference below) was used to solve the structure and we obtained a model that comprised of one hetero dimer in the asymmetric unit (ASU) with the log-likelihood gain (LLG) and the translation function Z score (TFZ) of 474 and 23 respectively. The corresponding Matthews coefficient (as published by Matthews in Molecular Biology, see full reference below) and solvent contents are $3.02 \text{ \AA}^3 \text{ Da}^{-1}$ and 59% respectively. The initial model was then subjected to AutoBuild (as published by Terwillinger et al in Acta crystallography, see full reference below) in PHENIX (as published by Adams et al in Acta crystallography, see full reference below) that provided a

partial model with $R_{\text{work}}/R_{\text{free}}$ of 31/41% for ~800 residues in 29 fragments with hetero dimeric core of PvcFRS. Subsequently, the model was manually built and completed by iterative cycles of building using COOT (as published by Emsley et al in *Acta crystallography*, see full reference below) and refinement using REFMAC (as published by *Murshudov et al in Acta crystallography, see full reference below*). After each cycles of manual building and refinement, the models were inspected and manually adjusted to correspond to the $2F_o-F_c$ and F_o-F_c electron density maps. During refinement, the ligand BRD1389 and Mg^{2+} ion were added based on positive peaks in difference Fourier maps and the model was subjected simulated annealing refinement using *phenix.refine* in PHENIX (as published by *Adams et al in Acta crystallography, see full reference below*). The final model was refined to 3.0 Å resolution with $R_{\text{work}}/R_{\text{free}}$ of 21.4/28.8%. The stereochemical quality of the model was analysed using MolProbity²³ and the model has good geometry quality and all residues are in favoured/allowed (92/8%) regions of the Ramachandran plot. The atomic coordinates of the PDB entry 7BY6 have been revised and the new statistics are updated in Supplementary Table 1. This revised text has been added to page 29.

Ref: Matthews, B. W. Solvent content of protein crystals. *J Mol Biol.* **33**, 491-497 (1968).

Ref: McCoy, A. J. *et al.* Phaser crystallographic software. *J. Appl. Crystallogr.* **40**, 658–674 (2007).

Ref: Adams, P. D. *et al.* PHENIX: A comprehensive Python-based system for macromolecular structure solution. *Acta Crystallogr. Sect. D Biol. Crystallogr.* **66**, 213–221 (2010).

Ref: Emsley, P. & Cowtan, K. Coot: Model-building tools for molecular graphics. *Acta Crystallogr. Sect. D Biol. Crystallogr.* **60**, 2126–2132 (2004).

Ref: Murshudov, G. N. *et al.* REFMAC5 for the refinement of macromolecular crystal structures. *Acta Crystallogr. Sect. D Biol. Crystallogr.* **67**, 355–367 (2011).

Ref: Terwilliger, T. C. *et al.* Decision-making in structure solution using Bayesian estimates of map quality: the PHENIX AutoSol wizard. *Acta Crystallogr. D Biol. Crystallogr.* **65**, 582-601 (2009).

7. A simulated anneal omit map identifies the binding site of BRD1389 which has patchy and non-connected electron density at 1 sigma. A best fit is shown in Figure 3d but is not unambiguous with regard to orientation of key elements of the compound. Taken together with overlays of other structures bound to their natural substrates (Fig 3c) and mutational resistance data, it shows the likely pose and orientation of the compound.

We agree with the reviewer's comment that the structural data together with conformational (Figure 2d, Supplementary Figure 8) and mutational resistance data (Figure 5c, 5d) unambiguously shows the orientation of BRD1389. In the revised paper we have shown the overlapping composite simulated annealed omit (SA-omit) and the final $2F_o-F_c$ map for active site region and it clearly shows electron density at 1 sigma level for the drug (Fig 2d, revised).

8. Figure 4 discusses "highly noticeable" loop openings and closing mediated by two key residues - Arg548 and His451. From the figures provided, the movement of Arg548 is clear but the histidines are less obvious and certainly do not appear to represent "major loop distortions". All 3 structures that are superimposed are of similar resolution and therefore, such loop areas but be subject to limitations of resolution as well as normal loop fluctuations within static structures.

We agree with the reviewers suggestion that the movement of His451 residue in the loop1 is less obvious and therefore we have removed “open and close conformation of histidine” remark from the text and in Figure 4b. However, structural comparison along the loop regions of the final model of *PvcFRS*-BRD1389 with *HsFRS*-Lphe (PDB-3L4G) (Figure 4) suggests contrasting conformations in the active site loops indicative of possible loop opening and closing. In addition, the observed electron densities in the present *PvcFRS*-BRD1389 structure clearly points to the differences in the conformations of the two active site loops (442-453 and 507-513) of *PvcFRS* when compared to *HsFRS* (Supplementary Figure 6b, 6c). Therefore, the structural comparison (Figure 4, revised) together with electron densities of loop region, added in Supplementary Figure 6 explain the major distortion of the loop upon ligand binding and a statement has been added in the revised main text on page 7.

9. After reading the section and inspecting the figure 4, I am still unsure what the take

home message is and if it is really as significant as the language used in parts would indicate. I wonder if this very long paragraph describing this section could be broken up and clarified to the reader, being wary of over-interpreting the structural data that is available.

We have broken up the paragraph describing the structural data based on key findings in the revised text on page_. The added subheadings are as follows:

- BRD1389 occupies the L-Phe site and an auxiliary site on *Plasmodium* cFRS. Page 6
- The active site of BRD1389-bound *Plasmodium* cFRS adopts unique conformations.

Page 7

- Structural basis of BRD1389 recognition by *PvcFRS*. Page 7
- Basis of selectivity and resistance-conferring mutations. Page 9

10. Subheadings would be of great benefit to this article. Not sure if they are allowed by the journal

but currently, there is little sign-posting of key findings making it difficult to follow (for what is a long article with many figure panels).

Subheadings have been added in the revised text as per the reviewer's suggestions.

Added subheadings are as follows:

- Introduction. Page 3
- Results
 - 1) BRD1389 binds cFRS selectively and inhibits aminoacylation via L-phe competition. Page 4
 - 2) Overall structure of *PvcFRS*. Page 5
 - 3) BRD1389 occupies the L-Phe site and an auxiliary site on *Plasmodium* cFRS. Page 6
 - 4) The active site of BRD1389-bound *Plasmodium* cFRS adopts unique conformations. Page 7
 - 5) Structural basis of BRD1389 recognition by *PvcFRS*. Page 7
 - 6) Basis of selectivity and resistance-conferring mutations. Page 9
- Discussion. Page 9

11. Figure 1i and j. Figure legend needs additional information. What is the orange and green in this panel? Is it possible to orient the purple and grey part of the structure in the same orientation as panels e, f, g & h as this would help the reader see the key points being made (which I think is the differences in the domain swaps in the formation of the tetramer)?

Detailed information for Figures 1i and 1j have been added in the legend as per reviewer's suggestion. The orientation of the tetrameric assembly of *PvcFRS* and *HsFRS* in Figure 1i and 1j has been revised to show the same orientation as other panels e-h as per reviewer's suggestions.

12. The SPR shown in Extended Data Figure 1 shows the comparison of *PfcFRS* vs *HscFRS*, with what appears to be a much slower off rates for *Pf* than *Hs*, and does not appear to return to baseline. Is ligand binding fully reversible? Or is there some evidence of long duration association or other interactions?

Yes, we can say that the drug binding is reversible as our enzyme assays show that the drug can be competed by L-Phe suggesting reversibility of drug binding.

Peer Review File

Reviewer comments, second round -

Reviewer #1 (Remarks to the Author):

The authors have addressed appropriately all the comments raised by this reviewer and the work is suitable for publication in Nature Communications.

Reviewer #2 (Remarks to the Author):

After a careful revision of the manuscript entitled "Structural basis of malaria parasite phenylalanine tRNA synthetase inhibition by bicyclic azetidines", it is clear that the authors considered the suggestions and addressed all the comments raised. As I could see, the manuscript has improved both in quality and readership and now it is suitable for publication in the Nature Communications.

Reviewer #3 (Remarks to the Author):

The authors have addressed my concerns regarding the structure refinement and conclusions. The correction to the asymmetric unit contents, additional explanation and figures for simulated anneal maps and inclusion of sub-headings makes the manuscript and related data is stronger and supports the conclusions made.

Please find below point-wise responses (in black) to reviewers queries/comments (in blue):

Revision I

Reviewers' comments:

Reviewer #1 (Remarks to the Author):

In the manuscript entitled "Structural basis of malaria parasite phenylalanine tRNA synthetase inhibition by bicyclic azetidines", the authors describe the mechanism of inhibition of a bicyclic azetidine toward *Plasmodium vivax* cytosolic phenylalanine-tRNA-synthetase (cFRS) and explore the structural aspects involved in the ligand-enzyme molecular recognition. The compound described in this study is a derivative of a previously reported series that proved to be highly active against different *Plasmodium* species, and effective against a *P. falciparum* murine model of malaria. Additionally, structural features that lead to selectivity over human cFRS and resistance to antimalarial agents are explored. The manuscript is the first to describe the structure of *Plasmodium vivax* phenylalanine-tRNA-synthetase in complex with an inhibitor, although with a resolution (3 Å) well above the desirable range. The authors additionally show that the binding mode of the compound provides a novel path to the design of selective molecules with respect to the human enzyme. The manuscript can be improved in several aspects.

COMMENTS and QUESTIONS

1. First, the validation of cFRS as a drug target should be briefly discussed.

In the revised paper we have added a line about cFRS as a validated drug target and also added the reference for the same. The added line reads – The *Pfc*FRS was validated both genetically and biochemically as a drug target for the bicyclic azetidine series of molecules as shown in our previous study. This has been added on page 3 of main text.

Ref: Kato, N. et al. Diversity-oriented synthesis yields novel multistage antimalarial inhibitors. *Nature* 538, 344-349 (2016).

2. Is there any controversy in the literature regarding targeting aminoacyl RNA synthetases in malaria drug discovery?

On the contrary, there are numerous laboratories worldwide that are now targeting aminoacyl RNA synthetases of malaria parasites for drug discovery. Even for this work we are aware of very stiff competition, and we are aware of advanced work on other targets within the aminoacyl tRNA synthetase family. Unrelatedly, tavaborole is an antifungal agent that inhibits leucyl-tRNA synthetase and is now a US FDA approved drug for treatment of onychomycosis. So, there is sufficient enthusiasm for drugging the malaria parasite encoded aminoacyl RNA synthetases. Please see few references that support targeting of aminoacyl RNA synthetases in malaria drug discovery:

1. Ref: Baragaña, B. et al. Lysyl-tRNA synthetase as a drug target in malaria and cryptosporidiosis. *Proc. Natl. Acad. Sci. USA*. **116**, 7015-7020 (2019).
2. Ref: Manickam, Y. et al. Drug targeting of one or more aminoacyl-tRNA synthetase in the malaria parasite *Plasmodium falciparum*. *Drug Discov. Today* **23**, 1233-1240 (2018).

3. Ref: Kato, N. et al. Diversity-oriented synthesis yields novel multistage antimalarial inhibitors. *Nature* 538, 344-349 (2016).
4. Ref: Herman, J. D. et al. The cytoplasmic prolyl-tRNA synthetase of the malaria parasite is a dual-stage target of febrifugine and its analogs. *Sci Transl Med.* 7, 288ra77 (2015)
5. Ref: Jain, V. et al. Structure of Prolyl-tRNA Synthetase-Halofuginone Complex Provides Basis for Development of Drugs against Malaria and Toxoplasmosis. *Structure* 23, 819-829 (2015).
6. Ref: Khan, S. et al. Structural basis of malaria parasite lysyl-tRNA synthetase inhibition by cladosporin. *J. Struct. Funct. Genomics* 15, 63-71 (2014).
7. Ref: Hoepfner, D. et al. Selective and specific inhibition of the plasmodium falciparum lysyl-tRNA synthetase by the fungal secondary metabolite cladosporin. *Cell Host Microbe.* 11, 654-663 (2012).

3. What has been done to validate genetically and chemically the enzyme as a molecular target? These aspects should be explored

Phenotypically, biochemically and via selection of drug-resistant mutations the *PfcFRS* has been validated as a drug target. As a matter of fact, this protein target has been awarded two GHIT grants (one very recently in 2020) for drug development against malaria parasites.

Ref: Kato, N. et al. Diversity-oriented synthesis yields novel multistage antimalarial inhibitors. *Nature* 538, 344-349 (2016).

4. Does cFRS meet druggability criteria for further drug discovery efforts? This should be clearly stated.

Indeed so. The cFRS meets the druggability criteria for further drug discovery as 1) bicyclic azetidine drugs inhibit plasmodium cFRS activity with high specificity (Kato et al 2016), 2) its 3-D structure is available now (via this paper) and this will set up further structure-based drug design, 3) cFRS is inhibited by bicyclic azetidine drugs with high potency (this and previous studies), 4) drug resistance mutations generated in parasites show the gene for cFRS as the target, and 5) this protein target has been awarded two GHIT grants (one in 2020) for drug development against malaria parasites. The text has been revised to include reviewer's suggestion on page 3 of main text.

5. The authors suggest that loop displacements occur because of ligand binding. However, there is the possibility of random loop fluctuations considering that these highly flexible elements are usually located at protein surfaces, and considering the limitations of x-ray crystallography. This point should be clarified.

We understand the reviewers concern. However presence of both loops (Loop1 and Loop2) in proximity of BRD1389 binding site together with structural comparison along the loop regions in *PvcFRS*-BRD1389 versus *HsFRS*-Lphe (PDB-3LAG) (Figure 4 is revised, Supplementary Figure 6b, 6c) suggests possible loop opening and closing. The electron density figures have been added as Supplementary Figure 6 and the text has been revised to highlight these points on page 7. The Figure 4 has been revised to include reviewers suggestion.

6. The sentence "A recent discovery of chemical series comprising a bicyclic azetidine

scaffold that exhibits multistage antimalarial activity, and can achieve single-dose cures in a mouse model of malaria" must be rephrased.

According to reviewer's suggestion, the above sentence is now rephrased as:

Recently, a series of small molecules based on a bicyclic azetidine core has been discovered that exhibits multistage antimalarial activity and can achieve single-dose cures in a mouse model of malaria. This has been changed on page 3.

Ref: Kato, N. et al. Diversity-oriented synthesis yields novel multistage antimalarial inhibitors. *Nature* 538, 344-349 (2016).

7. Were control compounds used in the enzymatic and binding assays?

Yes, we used appropriate controls in all assays. For enzymatic assay, recombinant maltose-binding protein (MBP) or no protein was used as controls (as published by *Sharma et al, see full reference below*). For binding studies ATP + L-Phe (1 mM each) were used as controls in place of the BRD drugs.

Ref: Sharma, A. & Sharma, A. Plasmodium falciparum mitochondria import tRNAs along with an active phenylalanyl-tRNA synthetase. *Biochem. J.* **465**, 459–469 (2015).

8. What was the *PfcFRS* solution concentration used for the crystallization experiments? Why *PfcFRS* structure has not been solved?

We have used $\sim 5 \text{ mg ml}^{-1}$ concentration for *PfcFRS* crystallization. We also have crystals of *PfcFRS* but the resolution is so far very limiting.

9. Further details on sequence alignment procedures should be added to the manuscript.

The details about the procedure and corresponding reference has been added in the revised text on page 29.

10. Considering that the study focuses on only one ligand, why HPLC runs for compound purity checking have not been carried out?

Indeed HPLC was carried out to check the purity of the drug used. We have added the HPLC details in our supplementary methods section and have added its profile in the Supplementary Figure 2.

Apart from the above highlighted comments, the manuscript brings new and relevant data for further efforts regarding malaria drug discovery and development.

We thank the reviewer for this positive response.

Reviewer #2 (Remarks to the Author):

The manuscript entitled “Structural basis of malaria parasite phenylalanine tRNA-synthetase inhibition by bicyclic azetidines” describes the structural and biochemical evidence that bicyclic azetidines are competitive inhibitors of L-Phe, required for the enzyme-catalyzed aminoacylation reaction that underpins protein synthesis in the parasite. Moreover, the manuscript indicates residue variations between the malaria parasite tRNA-synthetase and the human homolog that underpin the highly selective enzyme inhibition

and parasite killing by bicyclic azetidines. The authors present appealing data from a combination of biochemical and crystallographic studies that revealed the molecular underpinnings of Plasmodium tRNA-synthetase inhibition by bicyclic azetidines. The use of Plasmodium vivax phenylalanine tRNA-synthetase as a surrogate for the Plasmodium falciparum homolog is well-justified. The crystallographic structure of the phenylalanine tRNA-synthetase in complex with the potent bicyclic azetidine representative inhibitor and the mutagenic and biochemical studies conducted lays a foundation for designing the next generation of phenylalanine tRNA-synthetase inhibitors as antimalarial drug candidates. The collection and refinement statistics are satisfactory. The phenylalanine tRNA-synthetase complex structure is of modest quality (refined to 3.0 angstrom). The number of amino acid residues analyzed for Ramachandran outliers, rotameric score, model fitting is no more than 50% of the total amino acids for alpha subunit, which skews the statistical analyses. However, the observed electronic density within the enzyme catalytic site of alpha subunit was of enough quality to unambiguously fit the ligand to experimental data, thereby indicating the binding mode of the bicyclic azetidine derivative to phenylalanine tRNA-synthetase. Based on that, the inhibitor occupies both the L-Phe site and an auxiliary pocket within phenylalanine tRNA-synthetase, which represents a novel dual-site malaria parasite tRNA-synthetase inhibitor. The technical quality of the research reported is valid and appropriate. The degree of novelty and originality is high, and the conclusions are adequately supported by the data presented. Taken together, the data presented provides an important advance in the investigation of phenylalanine tRNA-synthetase as an attractive and validated molecular target for the discovery of new antimalarial drugs and, potentially, other diseases caused by apicomplexans, including toxoplasmosis and cryptosporidiosis. Therefore, the work will be interesting for the antimalarial community as new antimalarial targets and inhibitors are needed to be clinically validated. So, I am happy to suggest it for

acceptance. Nevertheless, there are some specific points that the authors should modify or reconsider previously to the publication of the work.

We thank the reviewer for this positive response. Indeed, our work has implications for toxoplasmosis and cryptosporidiosis. Moreover, the structure determination procedure has now been elaborated further and revised text has been added as shown below in the methods section on page 29.

We have additionally carried out crystal packing analysis of *Pv*FRS-BRD1389 complex and found no room for the missing N-terminal domain of the alpha subunit (residues 1-270). A crystal packing figure (Supplementary Figure 4a) has been added in the supplementary section. Further, we carried out SDS-PAGE analysis of *Pvc*FRS protein to determine the actual size of both the subunits of the protein in the crystal lattice and we found two bands at ~71.4 kDa and ~40 kDa (Supplementary Figure 4b). These two band sizes corresponds to the full-length beta subunit (618 residues, ~71.4 kDa) and a shortened alpha subunit, catalytic domain (residues 271-569, ~40 kDa) respectively. Thus, both crystal packing analysis and SDS-PAGE clearly suggest that the crystallized size of the alpha subunit is ~40 kDa (Supplementary Figure 4b). Hence, there are no missing 270 residues of the alpha subunit to account for in the crystal structure analysis. This information has been added in the revised main text on page 5-6 and both crystal packing and SDS-PAGE gel pictures have been added in the supplementary section (Supplementary Figure 4). The atomic coordinates of the PDB entry 7BY6 have been revised and the sequence information of the alpha subunit has been updated. The updated validation report is attached.

COMMENTS and QUESTIONS

1. Given the higher potency of BRD1389 against the whole parasite (Dd2 growth inhibitory activity of < 1 nM) related to the evaluated potency against the isolated *Pf* phenylalanine tRNA-synthetase (IC₅₀ = 12nM), the authors should comment whether a secondary mechanism of action is expected for the bicyclic azetidines.

Based on selection of parasite mutants that were partially resistant to cFRS (Kato *et al*, 2016), it seems that so far the only validated target of bicyclic azetidines is cFRS. We do not wish to speculate more on this score however it is noteworthy that:

- (1) The cFRS has been validated as a drug target both genetically and biochemically for bicyclic azetidine molecules as shown in our previous published work (Kato *et al*, 2016).
- (2) Also, we have shown very good correlation using bicyclic azetidine drugs for EC₅₀ and IC₅₀ values in our pervious published work (Kato *et al*, 2016).

Ref: Kato, N. et al. Diversity-oriented synthesis yields novel multistage antimalarial inhibitors. *Nature* 538, 344-349 (2016).

2. Page 11. The authors should add the standard deviation for data in Table 1, Extended Data 1C, and 1D.

Where possible standard deviations have been added and the figures have been revised . We have also included additional growth inhibition data (from different *Plasmodium* strains).

Reviewer #3 (Remarks to the Author):

In 2016, bicyclic azetidine compounds targeting phenylalanyl-tRNA synthetase (cFRS) were described as attractive new antimalarials. The current study describes the X-ray crystal structure of *Plasmodium vivax* cFRS bound to one of the compound series (BRD1389) and identifies key regions of the new drug target important for inhibitor binding. The inhibitory kinetics of the compound are characterised (activity against parasites is previously reported), and structural basis for selectivity against human homologs investigated. The data reported is new and will be of interest to the future design of antimalarial agents targeting cFRS, as well as potentially other anti-parasitic drug targets. The new structure identifies the binding site of these compounds and articulates the mechanism of inhibition. Key differences that provide selectivity between the human and plasmodium enzymes are identified, and residues contributing to compound binding and/or stabilisation are also identified. The structure of the vivax enzyme was solved via molecular replacement using the human homolog coordinates (3L4G.pdb). The human structure has a similar resolution (3.3 Ang to 3.0 as described for *PvcFRS*) and overall structure. Key differences in quaternary structure are described.

COMMENTS and QUESTIONS

1. The data obtained has a moderate resolution (3.0 Ang reported), and a structure with one heterodimer in the au, however, only ~ 50% of one monomer is modelled (Chain A = 296 residues or approx 50% of the chain, Chain B = 602 residues). The text states that there was no density observed for the N-terminal DNA binding domain of chain A. Overall refinement statistics show an Rfree of 28.2, which is a little surprising given the resolution

is limited to 3.0 Ang and a significant proportion of asymmetric unit remains unmodelled (~25% of au has no modelled density).

We have additionally carried out crystal packing analysis of *Pv*FRS-BRD1389 complex and found no room for the missing N-terminal domain of the alpha subunit (residues 1-270). A crystal packing figure (Supplementary Figure 4a) has been added in the extended/supplementary section. Further, we carried out SDS-PAGE analysis of *Pvc*FRS protein to determine the actual size of both the subunits of the protein in the crystal lattice and we found two bands at ~71.4 kDa and ~40 kDa (Supplementary Figure 4b). These two band sizes corresponds to the full-length beta subunit (618 residues, ~71.4 kDa) and a shortened alpha subunit, catalytic domain (residues 271-569, ~40 kDa) respectively. Thus, both crystal packing analysis and SDS-PAGE clearly suggest that the crystallized size of the alpha subunit is ~40 kDa (Supplementary Figure 4). Hence, there are no missing 270 residues of the alpha subunit to account for in the crystal structure analysis.

This proteolysis information has been added in the revised main text on page 5-6 and both crystal packing and SDS-PAGE gel pictures have been added in the supplementary section (Supplementary Figure 4). The atomic coordinates of the PDB entry 7BY6 have been revised and the sequence information of the alpha subunit has been updated. The updated validation report is attached.

2. Investigation of the MR probe, 3L4G, shows similar refinement statistics.

Molprobit analysis of 3L4G identifies significant causes for concern with regard to the quality of this structure (rated only in the 47th percentile of structures at this resolution). Therefore, I fear that 3L4G may be over-refined for its data resolution and this bias may be present in the new vivax structure.

We have addressed this concern. Please note that the deposited *HsFRS* (PDB ID: 3L4G) structure has 16 chains (A to P, eight α subunit and eight β subunit) and has 4 tetrameric assembly built of two $\alpha\beta$ heterodimers per asymmetric unit (ASU). In it, the N-terminal stretch of 1-180 residues fold into DNA binding domain. Amongst the eight α subunit in the ASU for *HsFRS*, only the C chain has this DNA binding fold portion. Therefore, we took two hetero dimers with or without DNA binding fold (CD and OP chains) of the *HsFRS* (PDB ID: 3L4G) as a template for molecular replacement solution of *PvFRS*-BRD1389 complex. The LLG and TFZ score for the obtained solution using CD template is 435 and 22 respectively, while the corresponding Matthews coefficient (as published by *Matthews in Molecular Biology, see full reference below*) along with solvent content are $2.24 \text{ \AA}^3 \text{ Da}^{-1}$ and 45% respectively. On the other hand, the LLG and TFZ score for the obtained solution using OP template is 474 and 23 respectively and the corresponding Matthews coefficient (as published by *Matthews in Molecular Biology, see full reference below*) and solvent content are $2.93 \text{ \AA}^3 \text{ Da}^{-1}$ and 58% respectively.

After the MR solution, the model was subjected to automatic model building module, AutoBuild in PHENIX that provided a partial model with ~ 800 residues in 29 fragments. Subsequently, the model was manually built and completed by iterative cycles of building using COOT) and refinement using REFMAC. After each cycles of manual building and refinement, the models were inspected and manually adjusted to correspond to the $2F_o-F_c$ and F_o-F_c electron density maps. During refinement, the ligand BRD1389 and Mg^{2+} ion were added based on positive peaks in difference Fourier maps and the model was subjected simulated annealing refinement using *phenix.refine* in PHENIX.

Furthermore, our present analysis of the stored enzyme and obtained crystals using SDS-PAGE gels clearly suggests that the N-terminal stretch of 1-270 residues has been cleaved during crystallization process (Supplementary Figure 4b). In addition, the crystal packing analysis also revealed that there is no room for the N-terminal domain of α subunit (Supplementary Figure 4b). Therefore, there is no possibility for the over refinement of the model. The representative regions of the final model with overlaid different electron density maps (Supplementary Figure 5 and 6) are now included in the revised supplementary section. These statements have been added in the revised text on page 29 and supporting figures were added in the revised supplementary section. The revised statistics of data collection and structure refinement are updated in Supplementary Table 1.

Ref: Matthews, B. W. Solvent content of protein crystals. *J Mol Biol.* **33**, 491-497 (1968).

3. Reading of the methods provided, does not suggest that any simulated annealing was performed post Phaser and prior to the initial build of this structure.

We apologize for not including these details in the original submission though we indeed used simulate annealing throughout the refinement process. An elaborated structure determination procedure now has been added in the revised text in the methods section on page 29. The atomic coordinates of the PDB entry 7BY6 have been revised and the sequence information of the α subunit has been updated. The updated validation report is attached here. The revised statistics of data collection and structure refinement are updated in Supplementary Table 1.

4. How was the MR model of 3L4G prepared for use as the probe?

The deposited *Hs*FRS (PDB ID: 3L4G) structure has 16 chains (A to P, eight α subunit and eight β subunit) and has 4 tetrameric assembly built of two $\alpha\beta$ heterodimers per asymmetric unit (ASU). In it, the N-terminal stretch of 1-180 residues fold into DNA binding domain. Amongst the eight α subunit in the ASU for *Hs*FRS, only the C chain has this DNA binding fold portion. Therefore, we took two hetero dimers with or without DNA binding fold (CD and OP chains) of the *Hs*FRS (PDB ID: 3L4G) as a template for molecular replacement solution of *Pvc*FRS-BRD1389 complex. The LLG and TFZ score for the obtained solution using CD template is 435 and 22 respectively, while the corresponding Matthews coefficient (as published by *Matthews in Molecular Biology, see full reference below*) along with solvent content are 2.24 $\text{\AA}^3 \text{Da}^{-1}$ and 45% respectively. On the other hand, the LLG and TFZ score for the obtained solution using OP template is 474 and 23 respectively and the corresponding Matthews coefficient (as published by *Matthews in Molecular Biology, see full reference below*) and solvent content are 2.93 $\text{\AA}^3 \text{Da}^{-1}$ and 58% respectively.

5. Figure 3d lists a simulated anneal omit map - did the authors inspect other areas of the structure (beyond the compound binding region) to inspect if this resulted in any changes?

In the revised paper, we have shown the overlapping composite simulated annealed omit (SA-omit) and the final $2F_o - F_c$ map for regions beyond active site – for example the connecting loop of the $\beta 1$ and $\beta 2$ sub-domains (Supplementary Figure 6a). We show the same for unique insertions in the $\beta 2$ sub-domains of the beta subunit of *Pvc*FRS-BRD1389 complex (Supplementary Figure 5). The electron density maps of the representative regions (Supplementary Figure 5) have been added in the revised supplementary section,

while the map for bound BRD1389 of the final model has been updated in Figure 2d. This has been revised in the main text of the manuscript on page 6.

6. Whilst the structure presented here has excellent quality scores, the R/Rfree indicate possible over-refinement (particularly with such disorder arising from absent domains).

Our new analysis shows that there is no over-refinement as the 1-270 residues of the alpha subunit are not present in the crystal. We below detail the full procedure for structure solution and refinement. There is no scope for over-refinement as simulated annealing was used regularly during the model building process. We have added representative maps to show the quality of the final model and all statistics are as per acceptable standards of PDB.

The initial model of the *Pv*FRS-BRD1389 complex was determined by molecular replacement (MR) method using the $\alpha\beta$ hetero dimer (OP chains) of the *Hs*FRS (PDB ID: 3L4G) as a template. The PHASER (as published by McCoy et al in crystallography, see full reference below) program in PHENIX (as published by Adams et al in Acta crystallography, see full reference below) was used to solve the structure and we obtained a model that comprised of one hetero dimer in the asymmetric unit (ASU) with the log-likelihood gain (LLG) and the translation function Z score (TFZ) of 474 and 23 respectively. The corresponding Matthews coefficient (as published by Matthews in Molecular Biology, see full reference below) and solvent contents are $3.02 \text{ \AA}^3 \text{ Da}^{-1}$ and 59% respectively. The initial model was then subjected to AutoBuild (as published by Terwillinger et al in Acta crystallography, see full reference below) in PHENIX (as published by Adams et al in Acta crystallography, see full reference below) that provided a

partial model with $R_{\text{work}}/R_{\text{free}}$ of 31/41% for ~800 residues in 29 fragments with hetero dimeric core of PvcFRS. Subsequently, the model was manually built and completed by iterative cycles of building using COOT (as published by Emsley et al in *Acta crystallography*, see full reference below) and refinement using REFMAC (as published by *Murshudov et al in Acta crystallography, see full reference below*). After each cycles of manual building and refinement, the models were inspected and manually adjusted to correspond to the $2F_o-F_c$ and F_o-F_c electron density maps. During refinement, the ligand BRD1389 and Mg^{2+} ion were added based on positive peaks in difference Fourier maps and the model was subjected simulated annealing refinement using *phenix.refine* in PHENIX (as published by *Adams et al in Acta crystallography, see full reference below*). The final model was refined to 3.0 Å resolution with $R_{\text{work}}/R_{\text{free}}$ of 21.4/28.8%. The stereochemical quality of the model was analysed using MolProbity²³ and the model has good geometry quality and all residues are in favoured/allowed (92/8%) regions of the Ramachandran plot. The atomic coordinates of the PDB entry 7BY6 have been revised and the new statistics are updated in Supplementary Table 1. This revised text has been added to page 29.

Ref: Matthews, B. W. Solvent content of protein crystals. *J Mol Biol.* **33**, 491-497 (1968).

Ref: McCoy, A. J. *et al.* Phaser crystallographic software. *J. Appl. Crystallogr.* **40**, 658–674 (2007).

Ref: Adams, P. D. *et al.* PHENIX: A comprehensive Python-based system for macromolecular structure solution. *Acta Crystallogr. Sect. D Biol. Crystallogr.* **66**, 213–221 (2010).

Ref: Emsley, P. & Cowtan, K. Coot: Model-building tools for molecular graphics. *Acta Crystallogr. Sect. D Biol. Crystallogr.* **60**, 2126–2132 (2004).

Ref: Murshudov, G. N. *et al.* REFMAC5 for the refinement of macromolecular crystal structures. *Acta Crystallogr. Sect. D Biol. Crystallogr.* **67**, 355–367 (2011).

Ref: Terwilliger, T. C. *et al.* Decision-making in structure solution using Bayesian estimates of map quality: the PHENIX AutoSol wizard. *Acta Crystallogr. D Biol. Crystallogr.* **65**, 582-601 (2009).

7. A simulated anneal omit map identifies the binding site of BRD1389 which has patchy and non-connected electron density at 1 sigma. A best fit is shown in Figure 3d but is not unambiguous with regard to orientation of key elements of the compound. Taken together with overlays of other structures bound to their natural substrates (Fig 3c) and mutational resistance data, it shows the likely pose and orientation of the compound.

We agree with the reviewer's comment that the structural data together with conformational (Figure 2d, Supplementary Figure 8) and mutational resistance data (Figure 5c, 5d) unambiguously shows the orientation of BRD1389. In the revised paper we have shown the overlapping composite simulated annealed omit (SA-omit) and the final $2F_o-F_c$ map for active site region and it clearly shows electron density at 1 sigma level for the drug (Fig 2d, revised).

8. Figure 4 discusses "highly noticeable" loop openings and closing mediated by two key residues - Arg548 and His451. From the figures provided, the movement of Arg548 is clear but the histidines are less obvious and certainly do not appear to represent "major loop distortions". All 3 structures that are superimposed are of similar resolution and therefore, such loop areas but be subject to limitations of resolution as well as normal loop fluctuations within static structures.

We agree with the reviewers suggestion that the movement of His451 residue in the loop1 is less obvious and therefore we have removed “open and close conformation of histidine” remark from the text and in Figure 4b. However, structural comparison along the loop regions of the final model of *PvcFRS*-BRD1389 with *HsFRS*-Lphe (PDB-3L4G) (Figure 4) suggests contrasting conformations in the active site loops indicative of possible loop opening and closing. In addition, the observed electron densities in the present *PvcFRS*-BRD1389 structure clearly points to the differences in the conformations of the two active site loops (442-453 and 507-513) of *PvcFRS* when compared to *HsFRS* (Supplementary Figure 6b, 6c). Therefore, the structural comparison (Figure 4, revised) together with electron densities of loop region, added in Supplementary Figure 6 explain the major distortion of the loop upon ligand binding and a statement has been added in the revised main text on page 7.

9. After reading the section and inspecting the figure 4, I am still unsure what the take

home message is and if it is really as significant as the language used in parts would indicate. I wonder if this very long paragraph describing this section could be broken up and clarified to the reader, being wary of over-interpreting the structural data that is available.

We have broken up the paragraph describing the structural data based on key findings in the revised text on page_. The added subheadings are as follows:

- BRD1389 occupies the L-Phe site and an auxiliary site on *Plasmodium* cFRS. Page 6
- The active site of BRD1389-bound *Plasmodium* cFRS adopts unique conformations.

Page 7

- Structural basis of BRD1389 recognition by *PvcFRS*. Page 7
- Basis of selectivity and resistance-conferring mutations. Page 9

10. Subheadings would be of great benefit to this article. Not sure if they are allowed by the journal

but currently, there is little sign-posting of key findings making it difficult to follow (for what is a long article with many figure panels).

Subheadings have been added in the revised text as per the reviewer's suggestions.

Added subheadings are as follows:

- Introduction. Page 3
- Results
 - 1) BRD1389 binds cFRS selectively and inhibits aminoacylation via L-phe competition. Page 4
 - 2) Overall structure of *PvcFRS*. Page 5
 - 3) BRD1389 occupies the L-Phe site and an auxiliary site on *Plasmodium* cFRS. Page 6
 - 4) The active site of BRD1389-bound *Plasmodium* cFRS adopts unique conformations. Page 7
 - 5) Structural basis of BRD1389 recognition by *PvcFRS*. Page 7
 - 6) Basis of selectivity and resistance-conferring mutations. Page 9
- Discussion. Page 9

11. Figure 1i and j. Figure legend needs additional information. What is the orange and green in this panel? Is it possible to orient the purple and grey part of the structure in the same orientation as panels e, f, g & h as this would help the reader see the key points being made (which I think is the differences in the domain swaps in the formation of the tetramer?)

Detailed information for Figures 1i and 1j have been added in the legend as per reviewer's suggestion. The orientation of the tetrameric assembly of *PvcFRS* and *HsFRS* in Figure 1i and 1j has been revised to show the same orientation as other panels e-h as per reviewer's suggestions.

12. The SPR shown in Extended Data Figure 1 shows the comparison of *PfcFRS* vs *HscFRS*, with what appears to be a much slower off rates for *Pf* than *Hs*, and does not appear to return to baseline. Is ligand binding fully reversible? Or is there some evidence of long duration association or other interactions?

Yes, we can say that the drug binding is reversible as our enzyme assays show that the drug can be competed by L-Phe suggesting reversibility of drug binding.

Revision II

Reviewers' comments:

Reviewer #1 (Remarks to the Author):

The authors have addressed appropriately all the comments raised by this reviewer and the work is suitable for publication in Nature Communications.

We thank the reviewer for this positive response.

Reviewer #2 (Remarks to the Author):

After a careful revision of the manuscript entitled “Structural basis of malaria parasite phenylalanine tRNA synthetase inhibition by bicyclic azetidines”, it is clear that the authors considered the suggestions and addressed all the comments raised. As I could see, the manuscript has improved both in quality and readership and now it is suitable for publication in the Nature Communications.

We thank the reviewer for this positive response. Indeed, the revised paper is significantly enhanced in terms of data content, scientific presentation and overall conclusions.

Reviewer #3 (Remarks to the Author):

The authors have addressed my concerns regarding the structure refinement and conclusions. The correction to the asymmetric unit contents, additional explanation and figures for simulated anneal maps and inclusion of sub-headings makes the manuscript and related data is stronger and supports the conclusions made. We thank the reviewer for this positive response. Indeed, the revised paper is significantly enhanced in terms of data content, scientific presentation and overall conclusions.